# Lower-skilled occupations face greater upskilling pressure in U.S. job ads

Di Tong[1], Lingfei Wu [2] & James A. Evans [3,4] ✉

Substantial scholarship has estimated the susceptibility of jobs to automation, but little has examined how job contents evolve in the information age as new technologies substitute for tasks, shifting required skills rather than eliminating entire jobs. Here we explore patterns of occupational skill change and characterize occupations and workers subject to the greatest re-skilling requirements in the United States. Recent work found that changing skill requirements are greatest for STEM occupations in the 2010s. Nevertheless, analyzing 167 million online job posts covering 721 occupations, we find that when accounting for distance between skills, skill change is greater for lower-skilled occupations: those with fewer skills, lower wages, and less educational requirements. We further investigate the differences in skill change across employer and market size, as well as social demographic groups. We find that jobs from small employers and markets experienced larger skill upgrades to catch up with the skill demands of their large employers and markets. While these varied skill changes could create uneven reskilling pressures across workers, they may also lead to a narrowing of gaps in job quality and prospects. We conclude by showcasing our model's potential to chart job evolution directions using skill embedding spaces.

With rapid advances in automation, computerization, and Artificial Intelligence (AI) technologies, substantial scholarship and public debate have focused on estimating the susceptibility of jobs to full or partial automation and the employment consequences that follow[1–8]. By contrast, relatively little work has examined how job contents evolve. New technologies often substitute for some skills within jobs while simultaneously creating new ones, which alter job skill demands and impose re-skilling pressure on workers[9–12]. Here we focus on the future of work not in terms of job elimination and displacement, but job transformation[11,13].

Measuring job transformation by skill change is a relatively recent and emerging research program. The bulk of prior empirical studies focus on analyzing specific jobs, skills and industrial forces of change[9,11,14–17]. This work demonstrates the importance of compositional skill change, but lacks quantification of the similarity between skills within jobs. This makes it difficult to systematically compare

jobs and understand which group of workers experiences more skill change. The major challenge for quantifying systemic job skill change lies in obtaining datasets that accurately represent job skill requirements[13]. Recent research has shifted from inferring skill change indirectly from shifts in relative wages and the supply of highly-educated labor[18], to direct measurement of skill composition. Still, this work relies on static, coarse skill taxonomies such as the Dictionary of Occupational Titles[14], the Occupational Information Network Database (O*NET)[13,19], or job titles from Census coding volumes[10]. Most recently, Deming and Noray (2020)[20] developed the first direct, dynamic and precise measurement of job skill change using high-quality job advertisement data. They quantify skill changes for a job over time by adding up the adjustments in the frequency or emphasis of all skills, including those that became outdated and new ones that were required. For the detailed equation, see Supplementary Note 2.1.

[1]Sloan School of Management, Massachusetts Institute of Technology, Cambridge, MA, USA. [2]School of Computing and Information, University of Pittsburgh, Pittsburgh, PA, USA. [3]Knowledge Lab and Department of Sociology, University of Chicago, Chicago, IL, USA. [4]Santa Fe Institute, Santa Fe, NM, USA. ✉e-mail: jevans@uchicago.edu

The limitation of this quantification, however, lies in its inadequate consideration of the graduated relationship between different skills. This oversight results in a skewed perspective, inducing an upward bias for technology-intensive and specialized occupations that possess many more listed skills than occupations with less specialization. For these higher-skilled and specialized occupations, when old skills are replaced by new ones, they are usually quite similar and likely not too difficult to learn, similar to how learning a new programming language is more straightforward for someone who already knows several. That measurement approach to skill change also exhibits downward bias for lower-skilled occupations with fewer skills, lower wages, and less educational requirements, as new skills needed for these occupations are fewer yet substantially different and more challenging to acquire. To overcome these limitations, we propose to investigate and quantify the distance between skills within the underlying space of complex human capital[21]. Some skills are closer to one another, creating local, accessible transition paths. This type of transition also implies less radical changes to the nature of the job, with smaller impacts on its returns and prospects. As illustrated by Fig. 1a, b, the transition required for a computer programmer to adapt to a shift in demand within the IT & Software Development skill cluster from one coding language to another is smaller than the required shift for a food batchmaker who manufactures large quantities of food for packaging and distribution to their first coding language (database administration).

Given this concern, we provide an alternative quantification method of skill change that uses embedding distance to model the space between skills. Deming and Noray (2020) suggested that higher-skilled STEM and technology-intensive occupations experienced the most skill content change from 2007 to 2019[20]. Here, we use the same dataset as Deming and Noray[20], but present a dramatically different observation when accounting for the distance between skills: lower-skilled occupations changed more than higher-skilled occupations, suggesting very different employment and skilling policies. Using the skill probability change measure as suggested by Deming and Noray[20], computer programmers change 175% more than food batchmaker (7.453 vs 2.708) (Fig. 1c). After modeling skill distance using skill vectors learned from direct and indirect co-occurrences of skills in job ads with word embedding[22], however, the assessment of who experiences the most radical skill change is reversed−food batchmaker changes 144% more than computer programmer (0.082 vs. 0.017 in cosine distance; Fig. 1d). We note that an intermediate version of Deming and Noray's (2020) measure, formally equivalent to their original metric[20] but using data-driven clusters also reverses their assessment (see Supplementary Note 2.3).

In this work, the Methods section explains how we quantify skill change by recovering skill distance with a machine learning model[22] that embeds skills in a high-resolution skill space. Based on this improved measurement, we analyze skill requirements from 721 occupations across 167 million job ads, which cover the near-universe of the U.S. labor market in the past decade. Our findings confirm that occupations categorized as lower-skilled encounter more radical skill upgrades. This holds true regardless of the criteria used for distinguishing between lower- and higher-skilled roles, whether by skill number, pay level, or required educational attainment (Fig. 2). To better understand skill change variations beyond lower- and higher-skilled jobs, we further compare occupational skill change across employer and market sizes. Our evidence suggests that small labor markets and small employers change more relative to their larger counterparts as they upgrade their skills and converge to the skill demand of large markets and employers[23] (Fig. 3). This catching-up pattern indicates that skill changes are more incremental at the frontier, where most high-skilled occupations, large employers, and markets are concentrated. In contrast, lower-skilled occupations and small employers and markets require larger leaps in the skill space as they

catch up. Additionally, we probe the implication of varying skill change across occupations on workers of different demographic groups. Female and nonwhite (African, Hispanic, and Asian American) workers experience more significant up-skilling requirements than male and white workers (Fig. 3). We conclude by discussing the broad potential of our skill space model, with a demonstrated application to illuminate job evolution directions (Supplementary Fig. S6).

## Results
### Lower-skilled occupations experience larger skill upgrades
Across the analyzed 721 occupations, lower-skilled occupations experience more skill change than higher-skilled occupations between 2010 and 2018. This holds no matter how we distinguish between lower- and higher-skilled occupations. The correlation between occupational skill change and logged average number of occupational core skills is −0.66 (Pearson $\rho$ with two-tailed test, $\rho(719) = -0.66$, $p < 0.001$, 95% CI [−0.71, −0.62].; Fig. 2a), between skill change and logged annual median pay for the occupation is −0.13 ($\rho$ with two-tailed test, $\rho(719) = -0.13$, $p < 0.001$, 95% CI [−0.20, −0.05]; Fig. 2b), and between skill change and average required years of entry-level education is -0.19 ($\rho$ with two-tailed test, $\rho(719) = -0.18$, $p < 0.001$, 95% CI [−0.25, −0.11]; Fig. 2c). In other words, re-skilling demand is more significant for workers in occupations requiring lower skill complexity and lower educational attainment with lower compensation. The only exception is that jobs requiring master's or doctorate degrees change more than those requiring bachelor's degrees, but both change markedly less than jobs requiring only an associate or high school degree Fig. 2c). In Fig. 2d, we sort twenty-two 2-digit SOC occupation categories by decreasing skill change to obtain face validity for the distinction between lower- and higher-skilled occupations. Occupations related to farming, fishing, and forestry, construction and extraction, and transportation and materials moving change most; while occupations related to management, business and financial operations, and computer and mathematical tasks change least. In Table 1 models 1−3, we regress occupational skill change on skill change, pay level, and years of required education. These models show that the association between occupation skill complexity and skill change is substantial and consistent when occupations are weighted by job frequency (average number of job posts in 2010 and 2018). This pattern holds when controlling for average concentration among occupation-level employers (domination of the market by one or a few employers) and within-occupation job skill heterogeneity and change (see Supplementary Note 3.6 and Supplementary Table S4). This pattern also manifests when we measure job skill level with O*NET job zones, which reflect both formal and informal training and experience (Supplementary Note 3.1).

Do lower-skilled job contents shift in an upskilling direction that suggests potential higher training costs as well as better job quality and prospects for workers? We show in Supplementary Fig. S8 that the substantial shift in skills observed within lower-skilled occupations signifies an upskilling trend rather than a deskilling one. The skill gap between higher- and lower-skilled occupations has narrowed from 2010 to 2018 ($t(720) = 20.50$ with two-tailed $t$ test of mean difference, $p < 0.001$, Cohen's d = 0.76, 95% CI [0.020, 0.025]; Supplementary Fig. S8a). These results indicate marked transformations in lower-skilled occupations that have aligned them more closely with their higher-skilled counterparts. This finding is consistent with the trend of rapid earnings growth among low-wage workers that have offset wage inequality in the last decade[24].

### Outsized skill upgrades required in small businesses and labor markets
Insofar as skill upgrading is essential, particularly for lower-skilled jobs, we now turn to explore a crucial question regarding the organizational and ecological environment of skill change: do jobs at smaller or larger

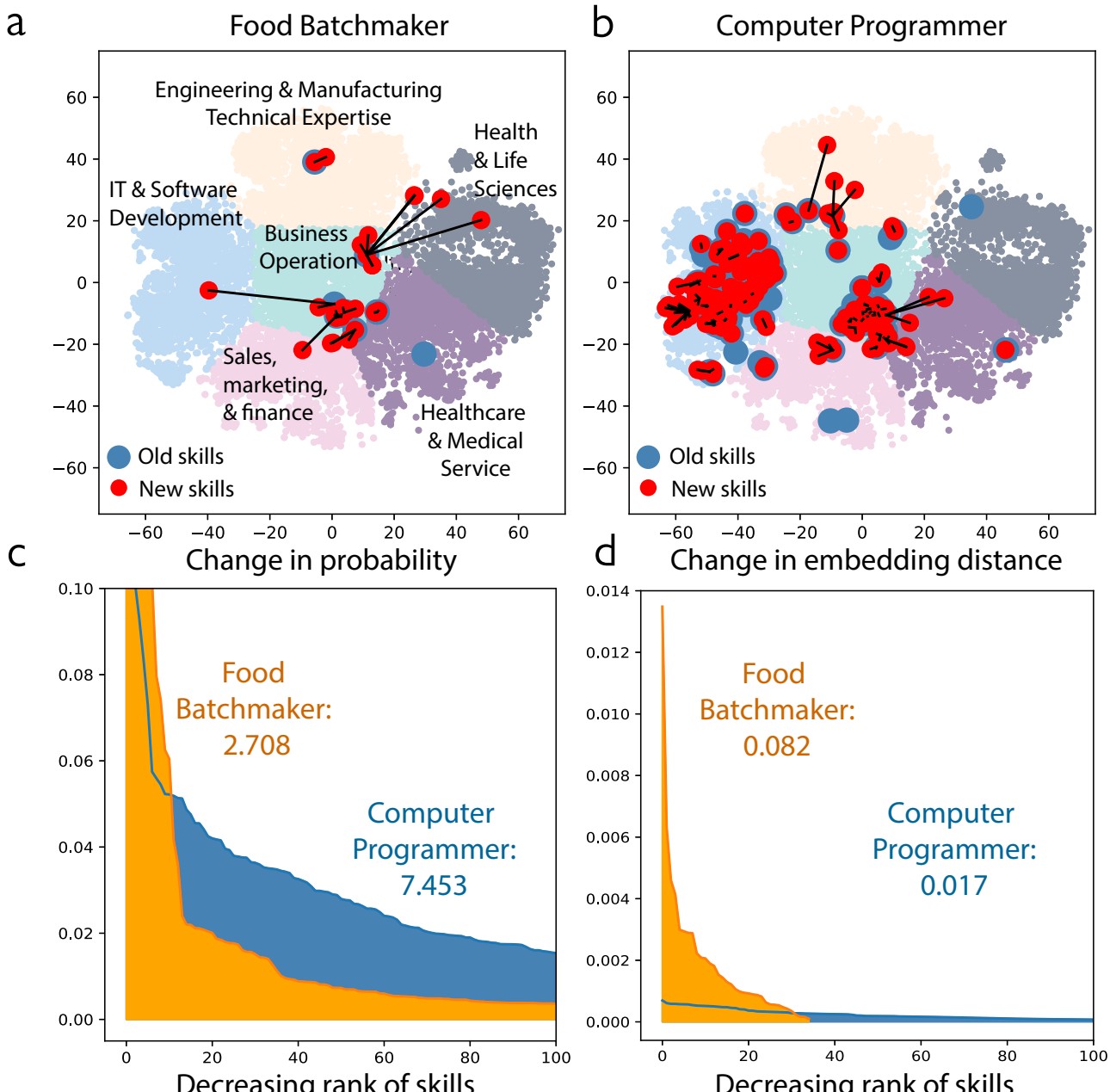

**Fig. 1 | Measuring occupational skill content change without considering skill proximity induces biases. a** Food Batchmakers have fewer new 2018 core skills (with top 5% occurrence) than computer programmers, but the new skills are on average located further from 2010 core skills. We illustrate the proximity and distance between skills by representing 15,182 skills in 6 well-formed clusters detected with the K-means algorithm from 2-dimensional t-SNE skill vectors transformed from the original 200-dimensional skill embedding. Each dot represents a 2-dimensional skill vector. Each new core skill (red dot) of food batchmakers is linked to its nearest 2010 old skill (blue dot), with the length of link proportional to the distance associated with skill transition. Compared with panel b, there are many fewer red dots in panel a, but the distance between the red dots and their nearest blue dots is typically larger. **b** Computer programmers have many new core skills in 2018 not among their 2010 core skills, but 2018 and 2010 skills are very similar.

**c** Without considering skill distance, computer programmers are assessed as experiencing substantially higher skill change than food batchmakers, 2010–2018. Each unit of the $x$ axis corresponds to a skill ranked from highest to lowest in terms of skill probability change. The $y$ axis denotes skill probability change for each skill (see Supplementary Information Section 2.2 for method details). The area under the orange and blue curves sum up the job-level skill change for food batchmaker and computer programmer, respectively (sums are reported in annotation text within the plot). **d** After controlling for skill distance, food batchmakers are associated with larger skill change than programmers. Each unit of the $x$-axis corresponds to a core skill ranked from highest to lowest in terms of their contribution to distance change within the job-level skill embedding. The $y$-axis denotes the distance change in skill embedding attributed to each skill (see Supplementary Information Section 2.2 for method details).

employers and local labor markets face greater pressure for upskilling? On one hand, research suggests that job roles in large firms and markets may change faster. These entities are often leading their fields at the skill frontier and possess the resources to attract and retain highly skilled workers[25–31]. These attributes may require large firms and

markets to engage in continuous learning that results in a faster pace of change in job skill requirements compared with smaller companies and markets. On the other hand, smaller firms and markets could pursue more significant skill upgrading to catch up with their larger counterparts. Theories of ecological inertia[32,33] and skill premia[34] also

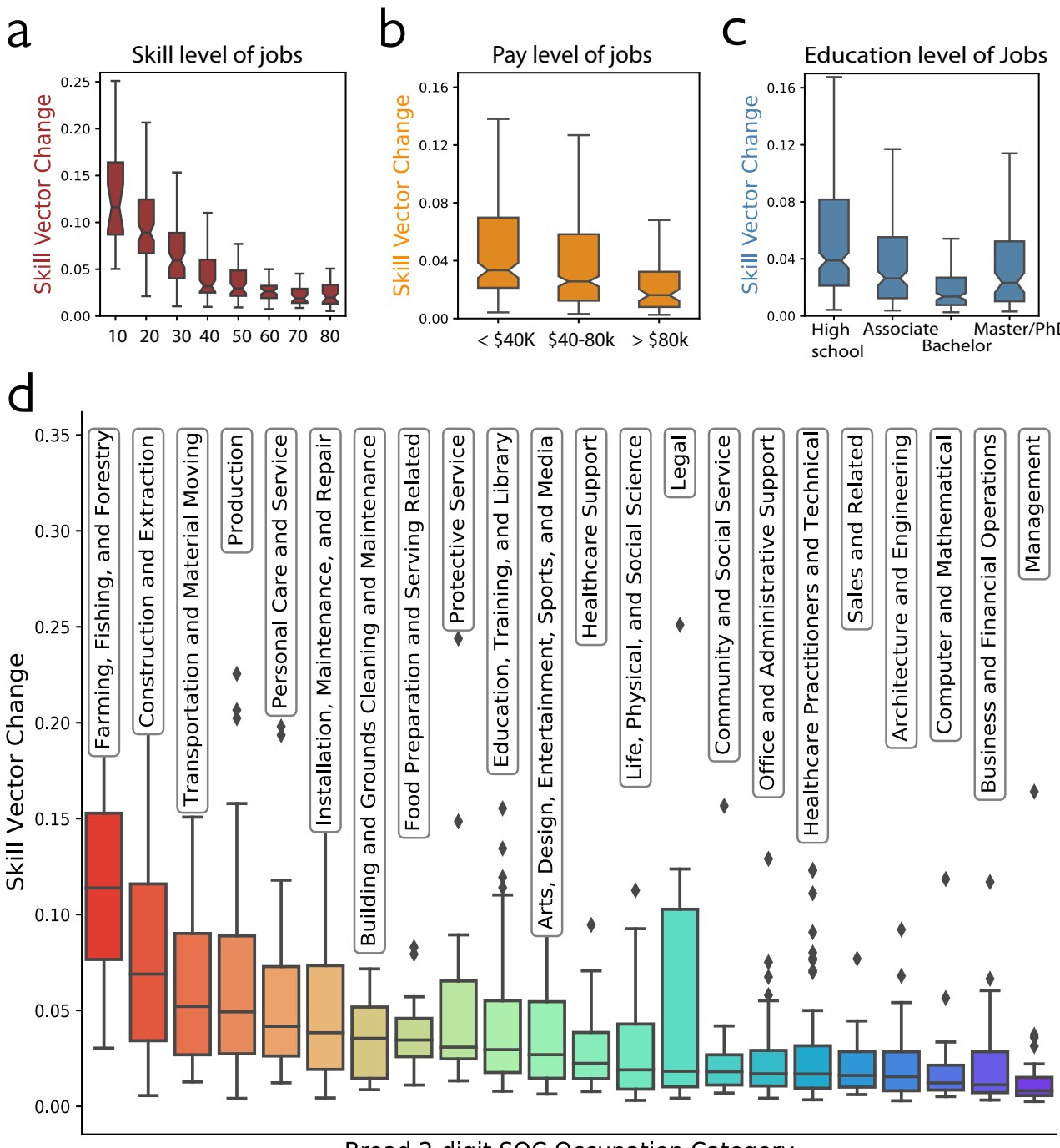

**Fig. 2 | Lower-skilled occupations experience larger skill change during 2010–2018 when skill distance is accounted for. a** Occupations with lower skill complexity have greater skill vector change. Each box blot denotes the distribution of skill vector change for occupations whose core skill numbers fall within the corresponding 10-point range (0–10, 10–20, ..., 70–80) shown on the x-axis (n = 484 occupations). For subplots (**a**–**c**), the center line indicates the median; the notches denote Tukey's approximate 95% confidence interval for the median; the box represents the interquartile range (25th–75th percentiles), and the whiskers extend to 1.5 times this range. **b** Occupations with lower pay have greater skill vector change. Each box plot denotes the distribution of skill vector change for occupations whose average annual median pay falls within the range labeled on the x-axis (n = 721 occupations). **c** Except for graduate degrees, occupations with lower education requirements change more; jobs requiring masters or doctorate degrees change more than those requiring bachelor degrees. Each box plot denotes the distribution of skill vector change for occupations whose average entry education requirement falls within the range labeled on the x-axis (n = 721 occupations). **d** The skill vector change distribution for 6-digit SOC occupations (n = 721) grouped by 2-digit SOC group categories. The box plots for occupation groups are ordered from largest to smallest median change. In each box plot, the center line indicates the median, the box represents the interquartile range (25th–75th percentiles), the whiskers extend to 1.5 times this range, and points beyond the whiskers are plotted as outliers.

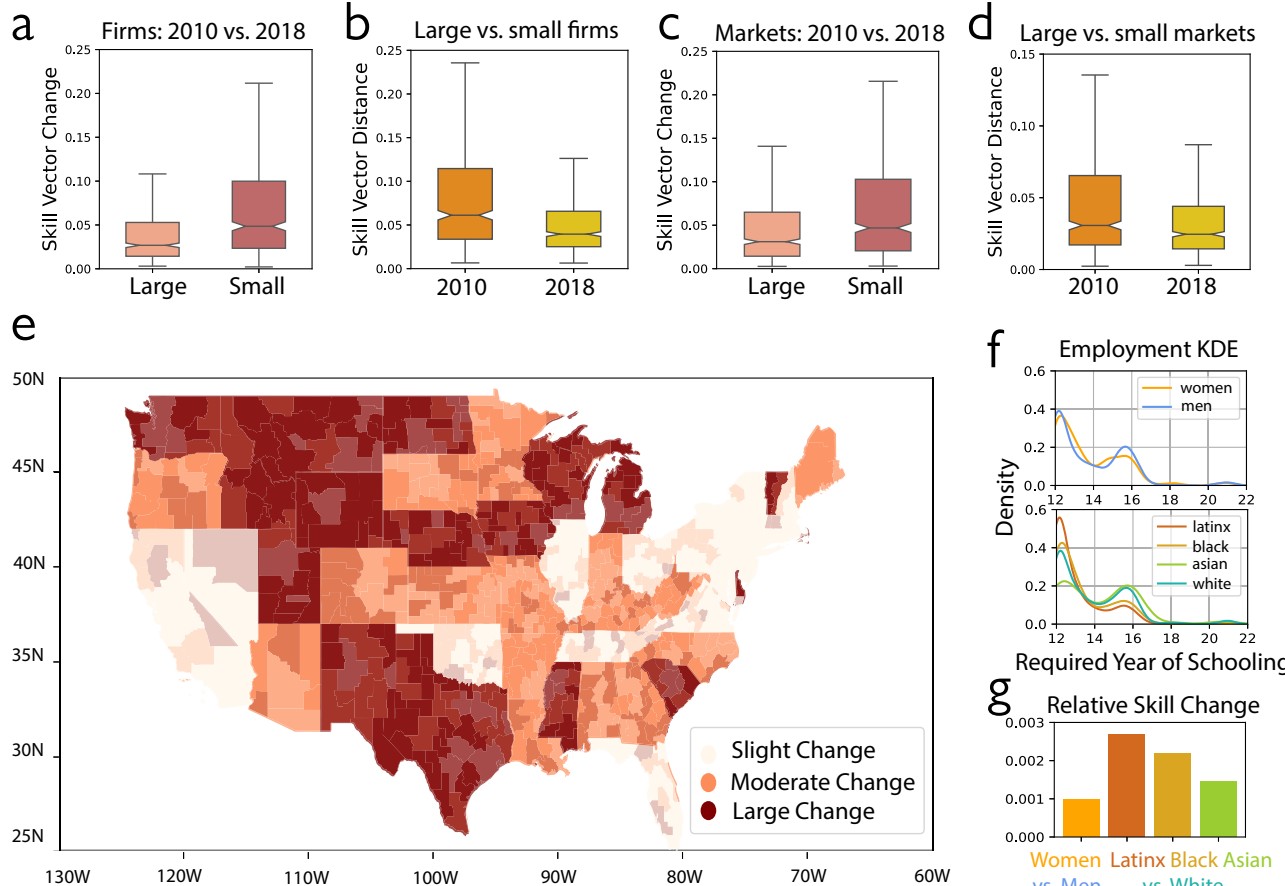

**Fig. 3 | Variation in skill demand change for different organizations, local markets, and social groups. a** Small firms have a higher level of occupational skill change than large firms, 2010–2018. The box plots denote the distribution of skill vector change for the overlapping occupations (*n* = 556 occupations) in large and small firms, respectively. For subplots (**a**–**d**), the center line indicates the median; the notches denote Tukey's -95% confidence interval for the median; the box represents the interquartile range (25–75th percentiles), and the whiskers extend to 1.5 times this range. **b** Small firms upgrade skill requirements and converge to large firm skill demands, 2010–2018. The box plots denote the distribution of occupational skill vector distance between large and small firms (*n* = 556 occupations) in 2010 and 2018, respectively. **c** Small markets have a higher level of occupational skill change than large firms, 2010–2018. The box plots denote the distribution of skill vector change for overlapping occupations (*n* = 625 occupations) in large and small markets, respectively. **d** Small markets upgrade skill requirements and converge to large markets' skill demands, 2010–2018. The box plots denote the distribution of occupational skill vector distance between large and small markets (*n* = 625 occupations) in 2010 and 2018, respectively. **e** Mapping occupational skill change in U.S. regions. States and commuting zones are grouped into three quantiles in terms of their average occupational skill change from low to high, labeled as slight, moderate, and large change. The average occupational skill change of each commuting zone is calculated as the sum of each occupation's skill change between 2010-2018 weighted by its 2018 local market share. State-level values are computed by averaging across commuting zones. **f** A larger proportion of male workers, relative to their total employment across occupations, are employed in higher-skilled occupations compared to female workers; A larger proportion of latinx and black workers, relative to their total employment across occupations, are employed in lower-skilled occupations compared to white and asian workers. Kernel Density Estimation (KDE) curves show the distribution of employment across occupations (*n* = 226) with varying required education years for six social groups. **g** Female workers and non-white workers face higher-reskilling pressure than male workers and white workers. The first bar represents the difference between female and male workers' weighted occupational skill change (*n* = 226 occupations). The other three bars denote the difference in weighted occupational skill change between three nonwhite groups and white workers (*n* = 226 occupations).

suggest that job roles in large companies and markets may change more slowly and gradually. As companies grow, they develop increased structural complexity that constrains radical change[35]. Moreover, with greater size and density of complementary skills, tasks, and occupations in large organizations and markets, ecological resilience may arise from the scale of network dependencies. This process may increase the ability of large companies and markets to manage uncertainty from their environment[32,33] and reduce the pressure for rapid shifts in job tasks[23,36–38].

To evaluate these dueling hypotheses, our study focuses on two aspects. First, we explore how the same occupation evolves within small versus large companies and markets over an eight-year observation period between 2010 and 2018. Second, we examine the development of a skill gap between large and small employers and

markets for the same occupations during this period. For the company-level analysis, we classify large employers as those with more than ten annual job postings in both starting and ending years of our analysis, placing them in the top 10% and distinguishing them from 90% of smaller employers. We computed two vector representations for the same occupation for each year, 2010–2018, based on the different skill requirements specified in large and small employers' job posts, for 556 overlapping occupations. This approach captures variations in core skill requirements from both employer size and time. With these vectors, we calculated the level of skill change for a given group of employers (e.g., large employers) as the average distance between their 2010 and 2018 occupation vectors. Additionally, we measured the skill demand gap between large and small employers in a given year (e.g., 2010) as the average distance between the vectors

**Table 1 | Skill change variation by occupation skill complexity, firm and market Size**

| | Occupation skill change | | | Firm-CZ-occupation skill change | | Firm-CZ skill change | |
|---|---|---|---|---|---|---|---|
| | (1) | (2) | (3) | (4) | (5) | (6) | (7) |
| Skill complexity | −0.012 | | | | | | |
| | (0.001) [p < 0.001] | | | | | | |
| Log annual pay | | −0.006 | | | | | |
| | | (0.001) [p < 0.001] | | | | | |
| Education | | | −0.001 | | | | |
| | | | (0.000) [p < 0.001] | | | | |
| Firm size | | | | −0.004 | | −0.001 | |
| | | | | (0.001) [p < 0.001] | | (0.000) [p = 0.009] | |
| CZ market size | | | | | −0.008 | | −0.003 |
| | | | | | (0.001) [p < 0.001] | | (0.001) [p < 0.001] |
| Constant | 0.077 | 0.086 | 0.034 | 0.177 | 0.240 | 0.115 | 0.144 |
| | (0.004) [p < 0.001] | (0.014) [p < 0.001] | (0.004) [p < 0.001] | (0.056) [p = 0.002] | (0.058) [p < 0.001] | (0.004) [p < 0.001] | (0.007) [p < 0.001] |
| Occ. FE | No | No | No | Yes | Yes | No | No |
| Observations | 721 | 721 | 721 | 4497 | 4497 | 13141 | 13141 |
| R² | 0.301 | 0.037 | 0.035 | 0.113 | 0.116 | 0.001 | 0.002 |
| Adjusted R² | 0.300 | 0.035 | 0.033 | 0.088 | 0.091 | 0.000 | 0.002 |
| Residual Std. Error | 0.045 (df = 719) | 0.057 (df = 719) | 0.057 (df = 719) | 0.084 (df = 4374) | 0.084 (df = 4374) | 0.109 (df = 13139) | 0.109 (df = 13139) |
| F Statistic | 309.416 (df = 1; 719) [p < 0.001] | 27.495 (df = 1; 719) [p < 0.001] | 25.901 (df = 1; 719) [p < 0.001] | 4.569 (df = 122; 4374) [p < 0.001] | 4.700 (df = 122; 4374) [p < 0.001] | 6.731 (df = 1; 13139) [p = 0.009] | 30.511 (df = 1; 13139) [p < 0.001] |

Significance of individual coefficients was determined using a two-sided *t*-test, and overall model significance was assessed with an *F*-test. *P*-values are reported without adjustment for multiple comparisons. CZ refers to the commuting zone. Skill complexity refers to the log occupational core skill number averaged over 2010 and 2018. Log annual pay refers to the log value of occupation average annual median pay. Education refers to the occupation average education year requirements. Firm and CZ market size are their corresponding log number of job posts averaged over 2010 and 2018. Models 1–3 use occupation as the unit of analysis and have occupational skill change from 2010 to 2018 as outcome variable. Models 4–5 use firm-CZ-occupation as the unit of analysis for units that are observed in both 2010 and 2018, and have firm-CZ-occupation skill change as the outcome variable. Models 6–7 use firm-CZ as the analytical unit and have firm-CZ average occupation skill change as the outcome variable. All regressions are weighted by the average number of job posts in 2010 and 2018.

representing the same occupations but constructed from large and small employers' job ads, respectively.

For the market-level analysis, we categorized job postings into large and small markets. We define local labor markets as commuting zones using 2010 data constructed by Penn State University based on the method developed by the Economic Research Service (ERS) of the U.S. Department of Agriculture[39,40]. Mapping the longitude and latitude locations of job posts to commuting zones, we classify a commuting zone as a large market if its annual job postings ranked within the top 10% among all 652 commuting zones in 2010 and 2018; otherwise, we considered it a small market. As in the firm-level analysis, we construct occupational skill vectors for large and small markets at different timepoints separately based on their different skill requirements for the same occupation in 2010 and 2018, for 625 overlapping occupations. With the same approach as for the small and large company comparison, we computed the average occupational skill change for large and small markets, as well as average occupational skill distance between large and small markets in 2010 and 2018, with very similar findings.

Occupational skill change and occupational skill gaps between small and large companies and markets reveal a strong skill convergence effect. Occupations in larger companies exhibit less skill change compared to smaller companies over time ($t(555) = −12.50$ with two-tailed test of mean difference, $p < 0.001$, Cohen's d = −0.53, 95% CI [−0.037, −0.027]; Fig. 3a). Similarly, occupations in larger markets exhibit less average skill change compared to their smaller counterparts ($t(624) = −11.90$ for two-tailed test, $p < 0.001$, Cohen's d = −0.48, 95% CI [−0.033, −0.024]; Fig. 3c). Additionally, the skill gap between

small and large companies and markets narrowed across the decade (for firm size, $t(555) = 12.11$ with two-tailed test of mean difference, $p < 0.001$, Cohen's d = 0.51, 95% CI [0.029, 0.040], Fig. 3b; for market size, $t(624) = 7.80$ with two-tailed test, $p < 0.001$, Cohen's d = 0.31, 95% CI [0.013, 0.022], Fig. 3d). These results suggest that between 2010 and 2018, a significant portion of skill change involved upgrading skills in smaller firms and markets with initially lower skill demands, resulting in small companies and markets undergoing more change and progressively resembling their larger counterparts. This pattern supports the catch-up hypothesis that skill change is more granular at the frontier, where most high-skilled occupations, large employers, and markets are located. In contrast, as lower-skilled occupations, smaller employers, and markets catch up, they must make more significant advances in skill levels to close the gap with their higher-skilled counterparts. In Fig. 3e and Supplementary Fig. S12, we graph variation in average occupational skill change on a U.S. map for large and small markets, highlighting that upskilling pressures are notably higher in rural regions compared to densely populated coastal states and urban areas. For our analysis of local exposure to upskilling pressure through the lens of automation risk, we link Lightcast occupations to automation-risk scores for 702 O*NET occupations from early work classifying "computerisable" jobs based on expert assessment[6] (see Supplementary Fig. S12).

Table 1, models 4−5 explain skill change between 2010 and 2018 within firm, commuting zone, and occupation level as a function of firm and market size. The coefficients of firm size and commuting zone market size suggest that smaller firms and markets have larger within-firm-commuting zone-occupation skill changes. This analysis can only

be performed on the subsample of occupations within firm and region that appeared in both 2010 and 2018, and so we also present regressions including occupations that disappeared or newly appeared in 2018 within each firm and commuting zone. Models 6 regresses average skill change within firms and commuting zones on firm size, and model 7 skill change within firms and commuting zones on market size. These models are consistent with the earlier pattern of results while accounting for changes in the composition of occupations within firms and markets. Supplementary Tables S5 and S6 in Supplementary Information demonstrate that more significant skill change in small markets and employers is robust after controlling for employer concentration (domination of market by one or a few employers) and changes of job skill heterogeneity within occupation (see Supplementary Note 3.6).

### Who faces the most upskilling demand?

To understand the implication of skill change on workers, we associate occupational skill change with worker demographics, leveraging statistics from the Bureau of Labor Statistics (BLS) based on the 2018 Current Population Survey (CPS). We measure the reskilling pressure faced by a social group as the weighted sum of occupational skill changes across all occupations, where the weights correspond to the proportion of the group's workers employed in each occupation relative to the group's total employment across occupations. In Fig. 3f, we present Kernel Density Estimation (KDE) curves illustrating each group's employment distribution across occupations of varying skill levels. Compared to female workers, a larger share of male workers are employed in higher-skilled occupations requiring college educational attainment or higher. A greater proportion of Latinx and Black workers, relative to their total employment, are employed in lower-skilled occupations requiring less than college educational attainment, compared to White and Asian workers. As lower-skilled occupations face greater reskilling pressure, social groups concentrated in these occupations are more vulnerable to these changes. In Fig. 3g, we compare weighted occupational skill changes between female and male workers and across racial groups. Results show that female workers and non-white workers experience higher reskilling pressure than male and white workers.

## Discussion

Our work advances the empirical science of skills and job change by developing a skill embedding model—skill2vec—that represents occupational skill content and reveals hidden and significant up-skilling demands. Using large-scale job advertisement data coupled with manifold learning methods from machine learning and artificial intelligence[22,41,42], we show that lower-skilled occupations experience greater skill change when accounting for the distance between skills. This finding directly challenges recent high-profile work[20] that concluded STEM and technology-intensive occupations face the most skill content change from 2007 to 2019.

This result underscores the value of modeling skill change through a continuous geometric representation. Our embedding distance model captures skill proximity through direct and indirect skill overlaps in jobs across the economy. Because jobs are held by skilled persons, and education and training are tuned to help persons obtain jobs, we conceptualize this skill distance in terms of the amount of cognitive and embodied expertise that must be altered for persons to shift from effectiveness in one job to another. In this way, skill distance reflects the training required to perform new skills and jobs. By introducing continuous geometry to supplant cluster and network topology, our model both captures skill interdependencies and generates vectors that permit the efficient and dependable calculation of distance between skills and jobs. Our continuous model allows for better quantification of re-skilling demand than discrete models at the occupational level[20] and enables us to explore variations in skill change

for workers across markets, employers, and demographic groups, providing a more nuanced understanding of the human capital system's complex structure.

From this skill space representation, we find that lower-skilled, less educated, and non-white workers who work in small companies and labor markets face greater upskilling demand from their jobs. They do not necessarily need to learn more new skills than their higher-skilled counterparts, but the new skills they must acquire to retain employment are more different from their initial skill set, and likely require more time, expense, and effort in retraining.

We further find a convergence pattern in skill demands. The skill convergence effect we document shows occupations in larger companies and markets exhibit less skill change compared to their smaller counterparts over time, with the skill gap between small and large entities progressively narrowing. This finding supports a catching-up hypothesis in which skill change is more granular at the frontier, where most higher-skilled occupations, large employers, and markets are concentrated. In contrast, lower-skilled occupations, small employers and markets require larger leaps in skill space as they catch up. This pattern aligns with theories of ecological inertia[32,33] and skill premia[34] suggesting that job roles in large companies and markets may change more slowly and gradually due to increased structural complexity that constrains it[35].

Our model also reveals the directional flow of skill change. By mapping occupational vectors, we observe nuanced movements at the human–machine interface (see Supplementary Note 2.7, Supplementary Fig. S6). For example, over the period of our study, card dealers added data skills, while power plant operators adopted environmental auditing. Given the variation in these directional shifts, systematic study is left to future work. We find technological change remains a powerful driver of skill transformation. IT-related skills increasingly diffuse across occupations. Prior work shows that machine-learning relevance predicts skill change[43]. While theory about skill-biased technological change suggested job loss for lower-skilled workers, recent evidence and our findings show how many of these workers instead upgrade their jobs[44,45], demonstrating that those in routine jobs experience an expansion and reconfiguration of their occupations rather than becoming unemployed[46].

Other macroeconomic forces also shape skill demand. Business cycles influence employer expectations. Recessions lead to increased demand for credentials, while tight markets lower entry requirements[47–49]. In Supplementary Note 3.5 and Supplementary Fig. S11, we show evidence consistent with prior research that upskilling slows during periods of low unemployment.

Our findings build on and extend prior approaches to measuring skill distance. Much recent work seeks to build a deeper empirical science of skills and job change through modeling skill distances. The expanding literature includes studies that either use labor flows to infer skill relatedness[50,51] or derive vectors from skill groups to calculate skill distance[43,52]. Nevertheless, such studies consider skills as largely independent from one another. To overcome this limitation, some have used dimensionality reduction techniques to encode skill relatedness with broad, orthogonal categories[53–56], while others have constructed skill networks to model skill interdependencies[21,37,57,58]. But factor analyses do not capture the relative distances between skill categories or between skills within the same category. Similarly, networks do not yield straightforward or continuous distances between skills or jobs, as many more and less paths may connect them.

Several limitations of our approach should be noted. First, the extent of necessary retraining depends on the individual worker's aptitude and preparation. Our data capture employer demand, not the actual cost borne by workers. In Supplementary Note 3.4, we explore this issue by estimating the education costs of re-skilling. Future work should incorporate data on learning costs—time, money, and stress—to measure re-skilling pressure more directly.

Second, job mobility and geographic constraints also shape re-skilling needs. Workers may avoid upskilling by relocating, though moves incur costs—transportation, housing, and lost social support. Our analysis does not capture these trade-offs, but they remain important for future work[59].

Third, our data capture shifts in hiring expectations, not current job content. Future research should use on-the-job data to more precisely identify pressure on current workers. Additionally, online job posting data may oversample certain types of positions and under-represent others, though our validation exercises post strong correlations with official labor statistics.

Despite these limitations, our results raise important implications for low-wage workers. As these workers upskill, they may perform their jobs more productively, which could lead to better compensation—especially if they gain bargaining power. Indeed, the wage gap between low- and median-wage jobs has narrowed in recent years[24]. Wage premiums at large over small firms have also declined[60]. Additionally, skill upgrades may turn dead-end jobs into stepping-stone opportunities that facilitate upward mobility[46]. Future work should examine how skill upgrades shape both compensation and career trajectories.

Our work identifies systemic inequities in the need for re-skilling. Workers in small firms, rural markets, and historically marginalized groups face disproportionate demands to adapt. If met successfully, these pressures may offer opportunities for mobility and improved job quality—but only with appropriate support. Policy responses must therefore attend to regional and organizational variation in how jobs evolve.

As teamwork increasingly dominates the workplace, future research can expand the unit of analysis beyond occupation for skill change and technology-skill complementarities. Large firms and markets—once pioneers of change—may now experience slower, more incremental evolution, while small firms and markets undergo sharper transformations. Future research should investigate whether scale always leads to inertia—or whether periods of surge and catch-up alternate over time. And how does this relate to the current emergence of AI-related skills across the workforce?

In sum, our methodological innovation of measuring skill change through embedding distances fundamentally reframes our understanding of who bears the burden of economic transformation and highlights the urgent need for policies that address these hidden inequalities in the evolving labor market.

## Methods
### Data
We analyzed a dataset of more than 167 million job ads from 2010 to 2018 provided by Lightcast. Lightcast parses, extracts, and codes multiple variables from each job ad using proprietary technologies that combine machine learning models with rules curated by in-house experts. Key variables include job title, 6-digit Standard Occupational Classification (SOC), geographic coordinates, required skills and education, salary, and employer name. Lightcast collects information from more than 40,000 job boards and company websites to create the largest dataset of the U.S. labor market[16]. Admittedly, not all new jobs appear online, but online recruitment represents an increasing share of labor market search, even for jobs historically associated with informal and offline recruitment. A 2013 study estimated 60–70% jobs were posted online[61]. Recent research suggested 85%[62]. To verify the representativeness of Lightcast data on the U.S. job market, we calculated occupational demand, pay level, and education requirements using Lightcast data and found that these values are highly correlated with BLS statistics in 2010 and 2018 (see Supplementary Note 1), justifying the overall consistency and credibility of Lightcast data during the time period of our analysis, despite coverage limitations[16]. Prior work based on Lightcast data has also conducted various validation exercises that compare Lightcast data with the Job Opening and Labor

Turnover Survey (JOLTS), CPS, Occupational Employment Statistics (OES), American Community Survey (ACS), and O*NET in terms of SOC occupation level job demand, employment, education, experience, and skill requirements[16,63].

### Training skill vectors
We apply the skip-gram word2vec model[22] to obtain the skill vector representations of jobs. Word2vec learns vector representation of words within a large-scale corpus such that two words frequently occurring in the same direct or indirect linguistic contexts remain close to one another within the latent space, often measured by cosine distance of the angle separating them along the surface[22] and suggesting semantic and/or syntactic correspondence. Using the 167 million job posts describing a set of required skills as instances of training context, we obtain a 200-dimensional vector for Lightcast's 15,182 unique skills. This embedding space encodes the similarity and distance between skills inferred from their direct and indirect co-presence across jobs. A direct co-occurrence between two skills manifests when they appear in a job advertisement together. A first-order indirect co-occurrence between two skills manifests when a third skill co-occurs with the first two skills, creating an indirect skill co-occurrence path. Further indirectness occurs when additional skills are required to connect the two skills along more distant pathways of co-occurrence. This association of directness and indirectness is akin to the direct and indirect pathways through which spectral algorithms underlying PageRank[64] and other Eigenvalue-based centrality scores[65] account for direct centrality, where other nodes connect to a focal node, and indirect centrality, where other nodes connect to nodes that themselves connect with the focal node. Direct co-occurrence disproportionally influences our assessments as the resulting neural embedding proximity approximates the integration of all co-occurrence pathways between skills, and direct co-occurrence represents a more probable pathway between two skills than indirect co-occurrence.

### Quantifying the magnitude of occupational skill change
We calculate the vector of a 6-digit SOC occupation $o$ at a given year $t$ by averaging the vectors of $m$ core skills required by $o$ in $t$. To derive reliable estimates, we focus on 721 active occupations with 100 or more job ads every year and analyze their 5% most frequently required skills in each year, referred to as core skills. In Supplementary Note 3.7, we demonstrate that our findings presented in the main text are robust to alternative definitions of skill content, including expansion of core skills to include all skills and a consideration of skill frequencies as weights. As core skills change over time, the same occupation may manifest evolving vector representations across years even though each skill has only a fixed, globally trained vector. In this way, we focus on occupational skill content change caused by the substitution of core skills[22]. Specifically, we use one minus the Cosine distance between vectors of the same occupation in 2010 and 2018 as the skill change score. We calculate job skill change for 721 occupations between 2010 and 2018. Calculated scores vary from 0.002 to 0.340, with a median of 0.027. As occupation vectors encode skill distance, this measurement of skill change reflects more precise efforts needed for re-skilling than prior work[20], as illustrated in Fig. 1.

Note that re-skilling costs inferred from an occupational skill change relative to that same occupation at a previous time point is a second-order measurement. This is fundamentally different from important first-order metrics that capture the absolute training cost required to qualify for an occupation from the position of no relevant skill, such as an occupation's educational requirement, O*NET job zone, or skill number. Our main empirical analysis examines how second-order re-skilling costs vary among occupations with different first-order skill levels (Fig. 2).

Finally, we explored an alternative measurement to characterize occupational skill change as its most significant distance of skill transition in Supplementary Note 3.2. This approach aims to correct the potential underestimation of skill change for higher-skilled vs. lower-skilled jobs under the scenario where a higher- and lower-skilled occupation both add the same number of new skills in 2018 that are extremely dissimilar to their 2010 skills, yet the higher-skilled occupation also adds many more similar skills that end up diluting the average change associated with the addition of dissimilar skills. With this alternative measurement, we examine individual skill transitions and focus on the largest skill transitions to rule out this possible bias. With this measurement, our conclusion—that lower-skilled jobs are more susceptible to change—remained consistent (see Supplementary Table S1).

### Validating skill and occupation vectors

In Supplementary Notes 2.4–2.6, we present several validation analyses for our estimated skill and occupation vectors. The t-SNE visualization of the skill and occupation vectors (Supplementary Figs. S3, 4) show that skills and occupations belonging to the same broader category form meaningful clusters (see Supplementary Note 2.4). Supplementary Fig. S5 further demonstrates that occupation skill vectors are consistently closer to their corresponding occupation title word vectors than to other random occupation word vectors.

As the skill embedding space trained on job posting data may skew towards overrepresented occupations and skills in online job space, we validate our embedding space with a large language model pre-trained on a broader corpus of text data from recent work[66]. *Labor Space* is derived from fine-tuning Google's BERT with representative descriptions of different levels of labor market entities from various corpora, including Occupational Information Network (O*NET) and European Skills, Competences, Qualifications, and Occupations (ESCO). In Supplementary Note 2.5, we show that relative distances between skills and between occupations in our skill2vec space are consistent with distances encoded in *Labor Space*.

Additionally, we validate the relationship between skill distance by analyzing its ability to predict worker mobility across occupations. To do this, we extracted data on the number of transitions between 1956 pairs of 6-digit SOC occupations in 2018 from CPS data[67], where individuals reported their occupation from the current and previous year. To assess the skill similarity between each pair of occupations, we calculated the cosine similarity using occupational skill vectors from our 2018 dataset. The Pearson correlation coefficient between skill similarity and logged worker mobility is 0.28 (Pearson $\rho$ with two-tailed test, $\rho(1954) = 0.28$, $p < 0.001$, 95% CI [0.24, 0.33]), suggesting that when workers shift jobs, they require similar skills. Second, we regress worker mobility against a baseline prediction based on occupation popularity (i.e., the logarithm of the product of employment in the two occupations using 2018 BLS data). This was done before and after accounting for skill vector distance in the regression. Our findings indicate that including skill similarity significantly enhances the baseline model, with the $R$-squared value increasing by 130% from 0.06 to 0.14.

We also compare our skill embedding approach to the more traditional factor analysis approach used in labor economics work to derive occupational distance measures[54,56]. In Supplementary Note 2.6, we demonstrate that the distances between occupations, as measured by skill factors, strongly align with those measured using skill embedding vectors. The two approaches also perform similarly in predicting worker mobility, with skill2vec explaining moderately more variance in occupation switches.

### Reporting summary

Further information on research design is available in the Nature Portfolio Reporting Summary linked to this article.

## Data availability

The raw job posting data used in this study are available under restricted access from LightCast; access can be obtained through a licensing agreement with LightCast, with details at https://lightcast.io/. The processed data generated in this study, including aggregated occupation-year level skill demands and skill embeddings derived from job postings, are available at https://github.com/di-Tong/SkillPaper/tree/master/IntermediateData (archived on Zenodo under https://doi.org/10.5281/zenodo.17444902). A comprehensive variable dictionary for LightCast data is provided in the Supplementary Information and at https://github.com/di-Tong/SkillPaper/tree/master/Codes. The publicly available datasets used in this study are available as follows: the 2010 Penn State University Labor-Sheds for Regional Analysis data at https://sites.psu.edu/psucz/data/, the 2018 CPS data at https://cps.ipums.org/cps/, O*NET job zone data at https://www.onetonline.org/, 2010 and 2018 BLS Occupational Employment Statistics (OES) data at https://www.bls.gov/oes/tables.htm, and Labor Force Statistics data at https://www.bls.gov/cps/cps_aa2018.htm. For further inquiries regarding data access, researchers may contact LightCast directly or reach out to the corresponding author.

## Code availability

The Python code used in the analysis is available at https://github.com/di-Tong/SkillPaper/tree/master/Codes and archived on Zenodo under https://doi.org/10.5281/zenodo.17444902.

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

## Acknowledgements

We thank Lawrence Katz, workshop participants at MIT Institute for Work and Employment Research, and conference participants at Labor and Employment Relations Association 2024 Annual Meeting for helpful comments. We also thank Bledi Taska and the staff at Lightcast for generously sharing their data and comments. J.E. thanks the NSF SBE-1829366, AFOSR FA9550-19-1-0354 and DARPA HR00111820006 for support, and L.W. acknowledges the support of Richard King Mellon Foundation, NSF SOS:DCI-2239418, and NIH R01GM164731.

## Author contributions

D.T., L.W., and J.A.E. jointly conceived and designed the study, contributed to data interpretation, and drafted, revised, and edited the manuscript. D.T. led the data analysis and implemented the models.

## Competing interests

J.E. maintains a commercial relationship with Google, which played no role in the design, implementation, or decision to publish the study. The other authors declare no competing interests.
