## [Transparent Peer Review file · Nature Communications]

Lower-skilled Occupations Face Greater Upskilling Pressure in U.S. Job Ads

Corresponding Author: Professor James Evans

Version 0:

Reviewer comments:

Reviewer #1

(Remarks to the Author)

I previously reviewed this manuscript for [Redacted]. I attach both the original review and the review for the resubmission. The authors have made some additional edits since, particularly with regard to more careful language and acknowledgment of actual training costs to attain new skills as an important lens through which to interpret their findings on skill distance. While I would still advocate for a tighter conceptual linkage between the subgroup analyses (Fig 3) and the main hypothesis—what's the narrative that rationalizes these patterns—the authors have addressed most of my comments. I think the current draft capably illustrates how alternatives measures of task space can change our understanding of which segments of the workforce have experienced greater levels of change in demanded skill.

Reviewer #2

(Remarks to the Author)

Review of "Low-Skilled Occupations Experience the Largest Skill Upgrades"

Summary:

This paper investigates the evolution of skill requirements within occupations in the United States between 2010 and 2018. Utilizing a large dataset of online job postings, the authors employ a novel skill embedding model, *skill2vec*, to quantify changes in skill demand. Notably, their findings challenge previous research by suggesting that low-skilled occupations experience the most substantial skill upgrades, as measured by the distance between old and newly required skills within the skill embedding space. Furthermore, they find that smaller firms and labor markets exhibit greater skill change compared to their larger counterparts, with non-white, male workers in low-skilled occupations facing the most significant upskilling demands.

Contribution:

The study addresses a crucial gap in the literature on the impact of new technologies on the labor market by explicitly focusing on the transformation of tasks within occupations and the relationship between skills. This approach offers a valuable contribution to the field of computational social science and provides a more nuanced understanding of how job content evolves in response to technological change.

Concerns and Suggestions:

While the dataset and overall approach are interesting, I have identified several areas that require further clarification and refinement to be published in Nature Communications:

1. Focus on IT Skill Diffusion vs. Occupational Upgrading:

The paper's central claim that low-skilled occupations experience the largest skill upgrades warrants further scrutiny. The presented evidence, particularly in Fig. S3, suggests a strong emphasis on the diffusion of IT-related skills across occupations, regardless of their initial skill level. For instance, the example of gaming dealers requiring big data processing skills might be better understood as a reflection of the increasing ubiquity of IT skills across various occupational domains.

Therefore, it could be argued that the findings primarily demonstrate a shift towards greater IT proficiency across occupations, rather than a genuine upgrading of occupation-specific skills within each occupation. The sorted list of 2-digit SOC codes in Fig. 2d could then be interpreted as reflecting the varying degree of proximity of different occupational categories to the IT domain.

While the observed diffusion of IT skills is an important empirical finding, it might be less groundbreaking and novel than a genuine demonstration of widespread upgrading of occupation-specific skills. I suggest the authors carefully reconsider their framing and delve deeper into the nature of the observed skill changes to determine whether they truly represent occupational upgrades or primarily reflect a broader trend of IT skill adoption.

2. Methodology and Validation of the Embedding Space:

The authors' choice of Word2vec for generating the skill embedding space raises questions regarding potential biases. Relying solely on job postings as training data could lead to an embedding space skewed towards overrepresented occupations and skills. This could result in clustering of dominant skills while less prevalent skills become scattered, potentially affecting the accuracy of skill distance calculations.

To mitigate this potential bias, I recommend exploring alternative approaches such as using pre-trained language models or fine-tuning existing models with a broader corpus of text data, as demonstrated in recent work [1].

Furthermore, the current validation of the embedding space appears insufficient for the standards of Nature Communications. I propose the following enhancements:

- * Visualization of the Embedding Space: Presenting a 2-dimensional visualization of the skill and occupation embedding space using techniques like t-SNE or UMAP would provide a valuable overview of the space's structure and allow readers to better grasp the relationships between skills and occupations.

- * Relocating Fig. 1(a) and (b) to the Embedding Space: Instead of relying on the co-occurrence network, which serves a primarily visual purpose, it would be more informative to redraw Fig. 1(a) and (b) within the actual skill embedding space used for the core analyses.

- * Thorough Validation of Occupational Vectors: The occupational vectors, obtained by averaging core skill vectors, require more rigorous validation. Possible analyses include:

- * Visualizing occupational vectors in the embedding space and color-coding them based on their 2-digit SOC codes to assess whether they cluster meaningfully according to their similarities.

- * Comparing the occupational vectors with static word vectors for occupations derived from the same embedding space. Demonstrating that occupation vectors derived from skill averages are consistently closer to their corresponding occupation word vectors than to other random occupation word vectors would provide further evidence of their validity.

Summary of Recommendations:

In summary, I believe this study presents a valuable approach to understanding occupational skill change. However, the current validation of the embedding space and the interpretation of the findings require critical refinement to meet the rigorous standards of Nature Communications. I strongly encourage the authors to address the concerns raised regarding potential biases in the embedding space and to provide a more comprehensive and nuanced discussion of the observed skill changes, particularly concerning the distinction between genuine occupational upgrades and the broader diffusion of IT-related skills. By strengthening the methodological rigor and refining their interpretation of the results, the authors can significantly enhance the impact and clarity of their contribution.

References:

[1] Kim, Seongwoon, Yong-Yeol Ahn, and Jaehyuk Park. "Labor Space: A Unifying Representation of the Labor Market via Large Language Models." Proceedings of the ACM on Web Conference 2024. 2024.

I hope these comments and suggestions are helpful in revising and strengthening your manuscript. I look forward to seeing the revised version of your work.

Reviewer #3

(Remarks to the Author)

Review of "Low-skilled occupations experience the largest skill upgrades"

Summary

This paper sets out to quantify how much occupations have changed in terms of their skill requirements between 2010 and 2018. To do so, it embeds skills in a high dimensional space using an approach from natural language processing (word2vec). The embedding captures information about how about 16k different skills are combined in the skill requirements of 167M job ads. The authors then use these embeddings to assess how entire occupations change. To do so, they average the embeddings of a set of core skills that are dominant in an occupation's job ads. Finally, they use this metric to explore which occupations changed the most over the period of their study. The main result shows that the use of skill embeddings overturns earlier findings that high-skill jobs had changed faster than low-skilled jobs. In fact, properly accounting for skill distances shows that the opposite holds: it is the low skill jobs that changed fastest. Furthermore, the authors show that jobs

change faster in smaller labor markets and smaller employers. Finally, low-skilled jobs that are predominantly held by non-white workers experienced particularly significant up-skilling requirements.

Review

The paper is part of a growing literature that tries to understand the nature of jobs in terms of highly disaggregated task or skill requirements. Its main contribution is that it quantifies how jobs change in a very large dataset of job ads, offering a high level of detail and wide, up-to-date coverage. Its main methodological novelty lies in using word2vec algorithms to reduce the dimensionality of the skill space. The relevance of doing so is immediately apparent, because it provides a diametrically opposed answer to an important question compared to existing approaches that count the number of skills that change in each occupation: it is not the high-skill jobs that changed most, but the low-skill ones. This has important implications not only for policy, but also for individuals' career planning and hiring decisions by companies. Overall, the paper is carefully executed and includes many important checks, for instance, about the representivity of the sample and the performance of the identified skill spaces in predicting job switches. Therefore, I am in general very positive about this work. However, I have some concerns and suggestions that I believe should be addressed. I list these below.

1) What are occupations? The authors ask by how much occupations change between 2010 and 2018. However, occupations are an artificial unit of observation that depends on the algorithm that the data provider used to classify jobs into occupations. To be sure, this problem is shared across the literature that relies on job ad data. However, given that the core research question is about within-occupation skill change it would be useful to reflect on the extent to which occupations are a useful unit of analysis. To explain better what I have in mind, there are two ways to think about dynamics on labor markets. One is careers, where workers move from one job to another. The other is vacancy chains, where jobs are moved from worker to worker. In the former, it is straightforward to determine how workers string jobs into a career. In the latter, it is much harder to decide which vacancies connect to the same underlying job and therewith to each other. The authors have data on vacancies, but are mostly interested in what happens to workers (for instance, which workers face the largest reskilling pressures). This leads to some complications.

First, even if job ads are unbiased reflections of vacancies, vacancies and jobs are not the same. For one, the vacancy-to-job ratio depends, among other things, on the turnover and average tenure in a job (jobs with much turnover will become vacant more frequently). This in turn is often related to average age and skill levels of workers. For another, it is not always clear whether the content of a job changes or whether an old job disappears and a new one appears. As a concrete example, consider the Gaming Dealers in Fig S3. In this occupation, skill requirements shifted from hospitality industry knowledge to big data processing. Arguably, however, this shift reflects two different jobs, one on the casino floor and the other in the IT department and it is unlikely that the same type of worker would be hired for these jobs or that a workers would transition between these task profiles (in fact, the old occupation's profile might have been closer to restaurant or hotel jobs than to the new occupation's profile). However, due to the data provider's classification, this is considered a within-occupation shift in skill requirements. To be sure, I don't think this paper is the place to resolve such ambiguities, but it would be helpful (1) to understand how job ads are classified into occupations by the data provider, (2) how the authors think about within versus across occupation skill changes and (3) reflect on the conceptual differences between job and vacancy data.

2) How does the paper's methodology compare to other work on occupational distance measures? In the introduction, the authors motivate their work in part with the following statement: "The limitation of [Deming and Noray's] quantification, however, lies in its inadequate consideration of the graduated relationship between different skills. This oversight results in a skewed perspective, inducing an upward bias for technology-intensive and specialized occupations that possess many more listed "skills" than occupations with less specialization." To analyze how occupations change over time, the authors quantify the distance between the occupations' old and new skill vectors. Whereas their focus on within-occupation skill requirements changes and the use of job ad data are rare, measures of skill distances have been widely used in labor economics (mostly, to study job switches). From this literature, the authors refer to the work by Gathmann and Schoenberg (2010 – Journal of Labor Economics). This paper shares the same problem as Deming and Noray's work that the authors are right to criticize: it assumes that all skills are equidistant. However, other papers in this literature address this problem in a similar way as the authors do, namely, using dimensionality reduction techniques, albeit more traditional ones, such as factor analysis. For instance, Ingram and Neumann (2006 – Labour Economics) use factor analysis on skills/tasks listed in the Dictionary of Occupational titles to study the returns to different skills. The same approach is used by Poletaev and Robinson (2008, Journal of Labor Economics) to measure skill distance in job switches. Finally, Robinson (2018, Journal of Human Resources) and Neffke et al. (2024, Research Policy) use factor analysis and then calculate weighted differences between these lower dimensional skill vectors, where weights reflect the contribution of each skill factor to wages or years of schooling requirements to take into consideration how valuable skills are (Robinson) or how long it would take workers to acquire the required additional skills in a job switch (Neffke et al.). Note that the latter question is closely related to the analysis in Fig. S7 and Table S3 that aims to connect skill differences to required reskilling efforts, as well as to remarks about reskilling in the discussion section. Furthermore, these papers carry out similar analyses to the ones in appendix 3.5, but more in-depth than the current paper.

This shows that there is quite a bit of work on embedding skills in lower dimensional spaces to capture the "graduated relationship between different skills." This does not mean that the use of more modern dimensionality reduction methodologies like word2vec is not valuable. However, to fully assess the value of these methods, one would have to compare them to existing efforts. In particular, I would have found it helpful to learn more about two issues. First, word2vec seems less transparent than factor analysis and it would be helpful to understand exactly what kind of information is used to measure how similar two skills are. Currently, the authors state: "This embedding space encodes the similarity and distance between skills inferred from their direct and indirect co-presence across jobs." What is conceptually the meaning of a direct co-occurrence? What of an indirect co-occurrence? In what sense are skills that co-occur directly or indirectly similar? How

much weight is put on either type of co-occurrence? If the last aspect is hard to quantify, it would already be helpful to answer the first questions to better understand what is going on under the hood. Second, it would be helpful to learn if word2vec based differences are more predictive of, for instance, job switches (as analyzed in the section titled: Validating Skill Vectors with Worker Mobility) than the factor-analysis based measures proposed in the labor literature. For instance, if factor analysis is used to reduce dimensionality how much of the variance in job switching does the Euclidian or cosine distance between two occupations in this space explain? Establishing that the authors' approach performs better would help its adoption in future work.

3) The regression analysis in Tables 5.4-5.7 strikes me as odd. This analysis tries to assess whether occupation change is more rapid in large or small employers and in large or small labor markets. While the question is interesting, I have major concerns about how the sample is created. The sample now consists, as far as I understood, of 488 occupations in four different contexts: (1) large employers (2) small employers (3) large labor markets (4) small labor markets, yielding a sample with $488 \times 4 = 1952$ observations. However, what is this supposed to be a sample of? After all, regression analysis is supposed to uncover the conditional expectation of a dependent variable as a function of a set of independent variables. The sample should be a reflection of some population from which units are drawn, ideally at random, such that this expectation can be estimated. However, what is the population of occupation-context cells supposed to reflect? This is conceptually hard to justify and in practice ignores that some cells are much larger (and therewith yield more accurate averages) than other cells.

Moreover, occupational changes in such cells may be due to actual changes of jobs, but also a shift of jobs between employers that prefer one version of an occupation to employers that prefer another. Note that the authors motivate the regression analysis as follows on page 4: "On one hand, research suggests that job roles in large firms and markets may change faster. These entities are often leading their fields at the skill frontier and possess the resources to attract and retain highly skilled workers. [...] On the other hand, smaller firms and markets could pursue more significant skill upgrading to catch up with their larger counterparts." Clearly, these hypotheses reflect considerations about changing jobs within a firm, but the regression analysis is not set up to capture such changes well and will mix within and cross-firm changes.

Instead, the authors could focus on more intuitive statistical units (e.g., firms). This is hard, because the unit of analysis would ideally be a job in a firm (see point 1 above), which requires connecting job ads in 2010 to job ads in 2018 within the same firm. Alternatively, one could study at the occupation-firm level and employ proper weighting to reflect that some occupations are larger than others. This still leaves a problem of appearing and disappearing occupations. However, I would consider this a feature of the world, not a problem of the methodology. To account for this, the authors could study how the firm changes as a whole (i.e., averaging skill vectors of all job ads related to the same firm, once in 2010 and once in 2018) and compare this to how occupations within firms change. A similar solution could exist for appearing and disappearing firms, whose skill vectors can be compared to one another. Note that adding information on the size of the labor market is straightforward, because either firms only have one location or the authors could split multilocation firms.

4) Minor comments

a. In the analysis of demographics, could the authors give more information on what kind of occupations they focus on? Typically, extreme overrepresentation is easier in small occupations than in larger ones. Moreover, it would be equally easy to just calculate weighted skill changes using a group's distribution across occupations as weights. In a sense, this would be a more accurate reflection of the types of skill changes different societal groups face. Does this change results?

b. In the discussion, the costs of relocation mentioned are transportation and moving costs. However, it is likely that the true costs of relocations are nonpecuniary, such as losses of the social capital and networks that support individuals both at a professional and a personal level.

c. I did not fully follow the following sentence on page 23 "To account for temporal compositional changes in job post number and skill number per job post, they weight the skill change rate calculated from equation (1) by multiplying the ratio of skill occurrence divided by post number in 2007 to that in 2019, for each occupation." Adding an equation for these weights would be helpful.

d. Notation: the subscript o in the righthand side of equation 6, page 26 turns into a variable (O) on the lefthand side, which is confusing.

e. How much variance is there within occupations in skill vectors? Are some occupations much more precisely defined than others? What would that mean?

Overall, I enjoyed reading the paper and I believe that its conclusions are sound. I hope these comments will be helpful to the authors.

Version 1:

Reviewer comments:

Reviewer #1

(Remarks to the Author)

The authors have satisfactorily addressed my previous comments.

(Remarks on code availability)

Reviewer #2

(Remarks to the Author)

The authors have thoroughly and thoughtfully addressed my initial concerns, substantially improving both the methodological rigor and interpretative clarity of the manuscript. They have revised their framing to provide a more nuanced discussion of the observed skill changes, appropriately distinguishing between IT skill diffusion and broader occupational upskilling. Furthermore, they implemented robust validation procedures for the embedding space, including comparisons with a state-of-the-art embedding model, visualization of skill and occupational vectors, and enhanced methodological transparency. I sincerely thank the authors for their careful and diligent revisions and for respectfully engaging with my comments. These thoughtful efforts have significantly strengthened the study. I believe the current version is well-prepared for publication in Nature Communications.

(Remarks on code availability)

Reviewer #3

(Remarks to the Author)

Review of revised version of “Low-skilled occupations experience the largest skill upgrades”

The authors substantially revised and expanded the initial version of the paper. In doing so, they addressed all my main concerns. I believe the current version is convincing and makes an important contribution to our understanding of changes in the US labor market in terms of the detailed skills that jobs require.

I have a few minor points that the authors should be able to address with little effort and that would amount to minor revisions:

1) Supplementary material, section 2.1.

a. The authors should define each of the variables in these equations.

b. Eq (1b) is σ -specific, but the right-hand side variables contain no subscript σ .

c. I think eq. (1b) is the (inverse of the) growth rate of the average number of skills per job in an occupation (after rearranging terms). Is that correct? Or should t_0 and t_1 be reversed? As far as I understand, this just loosely weights occupations that grow in complexity (i.e., that add skills) less heavily than those that don't but it is no “principled” correction against some kind of null model, correct? Given that the authors need to replicate the paper that motivated their analysis and thus “inherit” this equation, I would not expect a justification for this correction (in fact, the original paper neither seems to give a clear definition nor explanation for these weights), but it would help to understand how we should think about them.

2) Supplementary material, section 2.6: wording. I would typically not interpret a Pearson correlation of .51 or .59 as high. These metrics are statistically associated, but far from the same. What is more important is that the R^2 in job transition predictions is higher for the word2vec embeddings, suggesting that the word2vec approach captures more relevant information.

3) Potential inconsistencies about the relation between unemployment rate and skill change.

a. In the main text, the authors claim “we show evidence consistent with prior research that lower employment rates are associated with smaller yearly average upskilling change, 2010 to 2018.” There, the authors discuss employment rates, which move opposite of unemployment rates.

b. In the SI, section 3.5, the authors claim: “We computed the year-on-year average occupational skill changes (e.g., 2010-2011, 2011-2012, etc.) across all occupations from 2010-2018 and found a positive correlation with the yearly civilian unemployment rate released by BLS: pearson coefficient 0.95 with p-value less than 0.005. Fig. S11 shows that the yearly average occupational skill changes decline with the yearly unemployment rate from 2010-2018. Given that our skill change measure largely captures upskilling directions, this pattern is consistent with the business cycle literature.” Here, the stated correlation is positive, suggesting that the correlation with employment rates should be negative (i.e., opposite of what the main text claims). In line with this, Fig. 11 suggests that skill comove with unemployment rates. Is there a typo in the main text?

4) Section 3.6 of the SI. The regression analysis is now much more interpretable. However, it seems that the authors use the number of ads in a cell as weights. That is, they use frequency weights. The correct weights would be analytic weights. Can the authors clarify which weights they use?

5) I found the discussion section hard to process, because it does not clearly group different types of arguments. Currently, the section mixes summary, limitations, new discussions of other papers and even some additional results to conclude with a somewhat generic call for policy based on the paper's results. I think the paper could be made more impactful by improving the structure and flow of this section, clearly separating findings, limitations and open questions.

(Remarks on code availability)

Version 2:

Reviewer comments:

Reviewer #3

(Remarks to the Author)

The paper makes a convincing and important contribution to the literature and the authors have addressed all the concerns I raised in the review rounds.

(Remarks on code availability)

Dear Reviewers,

We sincerely thank you for these critical insights, suggestions, and additional research design ideas. This has helped us better convey the conceptual significance of our findings and provide stronger evidence for our conclusions regarding the character of skills change. We believe that addressing them with substantial new data, models, and arguments has dramatically strengthened the scope and clarity of the argument. Specifically, in response to reviewer concerns and recommendations, we made the following changes:

Major changes:

1. In response to Reviewer#2's request for more embedding validation, we conducted the following validation analyses as suggested by reviewer#2:
 - a. Validated the skill2vec embeddings with pre-trained LLM from Reviewer#2's recommended reference.
 - b. Visualized skill vectors in the embedding space and color-coded them with skill cluster family labels classified by the data vendor (lightcast).
 - c. Visualized occupation vectors in the embedding space and color-coded them based on their 2-digit SOC codes.
 - d. Compared our occupational vectors with static word vectors for occupations derived from the same embedding space.
 - e. Replaced the network-based illustration in Figure 1 with k -means cluster-based illustration using t-SNE transformation of actual skill embeddings.
2. In response to Reviewer#2's concern about the framing competition between IT skill diffusion vs. occupational upgrading, we revised figure S6 to de-emphasize IT skill diffusion pattern and clarified the framing in the paper.
3. In response to Reviewer#3's doubt about occupation as a meaningful unit of analysis, we:
 - a. Calculated the percent of job transitions across occupation in a year, to show that occupation is a meaningful analytical unit as workers tend to stay in the same occupation over time.
 - b. Developed a measure to capture the extent to which an occupation incorporates new types of job roles over time and ran the regressions controlling for this measure to show that job roles change do not fully explain the skill change.
4. In response to Reviewer#3's question about comparative advantage of skill embedding over traditional dimension reduction method, we conducted factor analysis to represent occupations with skill factors to validate the skill2vec embedding and compared the performance of factors and embedding in explaining the variance in job transitions.
5. In response to Reviewer#3's concern about ambiguous statistical units in regressions, we:
 - a. Added new regressions with an occupations as an analytical unit, each occupation is weighted by the corresponding number of job posts to adjust for the fact that

some occupation cells are larger and have more accurate average (Table S4 and S7).

- b. Added new regressions with a firm-commuting zone-occupation as an analytical unit to track within-firm-job changes, using a sample of units that appear in both 2010 and 2018, each firm-commuting zone-occupation is weighted by the corresponding number of job posts (Table S5 and S8).
 - c. Added new regression with a firm-commuting zone as an analytical unit to include occupations that disappeared and appeared in 2018, using a sample of units that appeared in both 2010 and 2018; each firm-commuting zone is weighted by the corresponding number of job posts (Table S6 and S9).
6. In response to Reviewer #1's request for tighter conceptual linkage between sub-group analysis and main finding and Reviewer #3's concern about the bias of approach used to assess social group exposure reskilling pressure, we:
- a. Revised Figure 3f to show that women, Latinx, and black workers are more disproportionately distributed in low-skilled occupations than in high-skilled occupations compared to men and white workers (new fig. 3f).
 - b. Added a new figure 3g, in which we replaced the original approach measuring a social group's skill change pressure with the more accurate approach suggested by Reviewer #3: the weighted sum of occupational skill changes across all occupations, where the weights correspond to the proportion of the group's workers employed in each occupation relative to the group's total employment across occupations.
7. Other changes: Added more clarifications of our data, method, and limitations; and revised framing to present a more coherent narrative, as suggested by the reviewers.

In sum, we believe that addressing these concerns and suggestions has significantly improved the manuscript. We believe these enhancements warrant a reconsideration of our paper's suitability for *Nature Communications*. Detailed responses are provided below, with original comments in blue and our responses in black. Thank you again for taking the time to engage with this research.

Reviewer 1

1.1. I previously reviewed this manuscript for [Redacted]. I attach both the original review and the review for the resubmission. The authors have made some additional edits since, particularly with regard to more careful language and acknowledgment of actual training costs to attain new skills as an important lens through which to interpret their findings on skill distance. While I would still advocate for tighter conceptual linkage between the subgroup analyses (Fig 3) and the main

hypothesis—what’s the narrative that rationalizes these patterns—the authors have addressed most of my comments. I think the current draft capably illustrates how alternatives measures of task space can change our understanding of which segments of the workforce have experienced greater levels of change in demanded skill.

R1.1. We are pleased that you understand our purpose and value the motivation and significance of our work, as well as your support for its publication. In response to your feedback, we have further strengthened the conceptual link between the subgroup analyses and the main finding with the following changes:

1. in the introduction and the section discussing subgroup analysis, we strengthens the emphasis of the “catching up” hypothesis that connects employer/market size analysis to the main finding: skill changes are more granular at the frontier, where most high-skilled occupations, large employers, and markets are concentrated; in contrast, low-skilled occupations, small employers, and markets require larger leaps in the skill space as they catch up;
2. we added fig.3f to show the employment distribution of different demographic groups across occupations of varying skill levels, motivating the demographic analysis as an exploration of the implication of the main finding on demographic groups’ susceptibility to skill change pressure.

We greatly appreciate your time and constructive insights.

Reviewer 2

2.1. Summary: This paper investigates the evolution of skill requirements within occupations in the United States between 2010 and 2018. Utilizing a large dataset of online job postings, the authors employ a novel skill embedding model, skill2vec, to quantify changes in skill demand. Notably, their findings challenge previous research by suggesting that low-skilled occupations experience the most substantial skill upgrades, as measured by the distance between old and newly required skills within the skill embedding space. Furthermore, they find that smaller firms and labor markets exhibit greater skill change compared to their larger counterparts, with non-white, male workers in low-skilled occupations facing the most significant upskilling demands.

Contribution: The study addresses a crucial gap in the literature on the impact of new technologies on the labor market by explicitly focusing on the transformation of tasks within occupations and the relationship between skills. This approach offers a valuable contribution to the field of computational social science and provides a more nuanced understanding of how job content evolves in response to technological change.

Concerns and Suggestions: While the dataset and overall approach are interesting, I have identified several areas that require further clarification and refinement to be published in Nature Communications:

R2.1. Thank you for recognizing the novelty and contribution of our study and the unique contribution of our skill2vec model in capturing skill transformation within the workforce. We appreciate your thoughtful and precise summary, which highlights how our findings challenge previous research and provide new insights into understanding how job content evolves in response to technological change. We take your comments as a strong endorsement of this work's potential for publication.

2.2. Focus on IT Skill Diffusion vs. Occupational Upgrading: The paper's central claim that low-skilled occupations experience the largest skill upgrades warrants further scrutiny. The presented evidence, particularly in Fig. S3, suggests a strong emphasis on the diffusion of IT-related skills across occupations, regardless of their initial skill level. For instance, the example of gaming dealers requiring big data processing skills might be better understood as a reflection of the increasing ubiquity of IT skills across various occupational domains. Therefore, it could be argued that the findings primarily demonstrate a shift towards greater IT proficiency across occupations, rather than a genuine upgrading of occupation-specific skills within each occupation. The sorted list of 2-digit SOC codes in Fig. 2d could then be interpreted as reflecting the varying degree of proximity of different occupational categories to the IT domain. While the observed diffusion of IT skills is an important empirical finding, it might be less groundbreaking and novel than a genuine demonstration of widespread upgrading of occupation-specific skills. I suggest the authors carefully reconsider their framing and delve deeper into the nature of the

observed skill changes to determine whether they truly represent occupational upgrades or primarily reflect a broader trend of IT skill adoption.

R2.2. Thank you for raising this important point about the framing. We consider many instances of IT skill diffusion as one type of upskilling. However, IT skill diffusion is definitely not the sole direction of skill upgrades observed. For instance, human-interface skill atoms, such as supply-chain management, environmental auditing, and rehabilitation therapy, are also rising in importance within the skill space, while some machine-interface skill atoms, like IT admin support systems, mainframe systems, and Java frameworks, are declining. While we showed examples of occupations moving to machine-interface tasks, there are also transitions towards human-interface tasks: e.g., purchasing agents transitioning from transaction processing skill atom to supply chain management skill atom; power plant operator moving from power tools operation atom to environmental auditing atom.

Our paper focuses on documenting the magnitude of skill change faced by different segments of the labor force and suggests the broad upskilling rather than deskilling direction of change with evidence of narrowing skill gaps between high- and low-skilled occupations and between large and small firms/markets from 2010 to 2018 (Fig.S8). We intend to use Fig. S6 (formerly Fig. S3) solely as a demonstration tool to explore exemplary directions of skill change, leaving a comprehensive and rigorous examination of the complex and multifaceted specific directions of skill demands for future work. We have revised fig. S6 to de-emphasize the IT skill diffusion pattern by including a broader range of examples. The updated figure now highlights both human- and machine-interface skill changes, showcasing re-skilling directions more comprehensively. Additionally, the discussion section has been revised to clarify that IT skill diffusion represents just one direction and a potential source of skill change, emphasizing the broader evidence of upskilling while acknowledging that specific re-skilling directions warrant further study in future research.

2.3 Methodology and Validation of the Embedding Space: The authors' choice of Word2vec for generating the skill embedding space raises questions regarding potential biases. Relying solely on job postings as training data could lead to an embedding space skewed towards overrepresented occupations and skills. This could result in clustering of dominant skills while less prevalent skills become scattered, potentially affecting the accuracy of skill distance calculations. To mitigate this potential bias, I recommend exploring alternative approaches such as using pre-trained language models or fine-tuning existing models with a broader corpus of text data, as demonstrated in recent work [1].

References:

[1] Kim, Seongwoon, Yong-Yeol Ahn, and Jaehyuk Park. "Labor Space: A Unifying Representation of the Labor Market via Large Language Models." *Proceedings of the ACM on Web Conference 2024*. 2024.

R2.3. We agree that the skill embedding space is biased toward job content more likely to appear online. We therefore have implemented an alternative approach to validate this space in Supplementary Information section 2.5 and mentioned in the methods section, based on the recommended reference below:

We represent occupations and skills in our dataset using *Labor Space* BERT embeddings from the recommended reference. We then compute pairwise cosine similarities between all occupation pairs and skill pairs using both the *Labor Space* and our skill2vec embedding representations. To evaluate the alignment between the two spaces, we compare the cosine similarity values obtained from each. For occupation pairs, the Pearson correlation between the two sets of pairwise similarities is 0.34 ($p = 0.00$), while for skill pairs, it is 0.29 ($p = 0.00$). As a reference, we generated a random baseline by reshuffling occupation and skill names in the *Labor Space* vectors before computing correlations. The resulting correlations are 0.00 ($p = 0.89$) for occupations and 0.00 ($p = 0.36$) for skills, confirming that the observed correlations are meaningful.

2.4 Furthermore, the current validation of the embedding space appears insufficient for the standards of Nature Communications. I propose the following enhancements:

R2.4. We are grateful for detailed suggestions. We have implemented all these suggestions as in Supplementary Information section 2.4 and mentioned in the methods section.

(1) Visualization of the Embedding Space: Presenting a 2-dimensional visualization of the skill and occupation embedding space using techniques like t-SNE or UMAP would provide a valuable overview of the space's structure and allow readers to better grasp the relationships between skills and occupations.

Done.

(2) Relocating Fig. 1(a) and (b) to the Embedding Space: Instead of relying on the co-occurrence network, which serves a primarily visual purpose, it would be more informative to redraw Fig. 1(a) and (b) within the actual skill embedding space used for the core analyses.

Done.

(3) Thorough Validation of Occupational Vectors: The occupational vectors, obtained by averaging core skill vectors, require more rigorous validation. Possible analyses include:

(3A) Visualizing occupational vectors in the embedding space and color-coding them based on their 2-digit SOC codes to assess whether they cluster meaningfully according to their similarities.

Done.

(3B) Comparing the occupational vectors with static word vectors for occupations derived from the same embedding space. Demonstrating that occupation vectors derived from skill averages are consistently closer to their corresponding occupation word vectors than to other random occupation word vectors would provide further evidence of their validity.

Done.

2.5 Summary of Recommendations: In summary, I believe this study presents a valuable approach to understanding occupational skill change. However, the current validation of the embedding space and the interpretation of the findings require critical refinement to meet the rigorous standards of Nature Communications. I strongly encourage the authors to address the concerns raised regarding potential biases in the embedding space and to provide a more comprehensive and nuanced discussion of the observed skill changes, particularly concerning the distinction between genuine occupational upgrades and the broader diffusion of IT-related skills. By strengthening the methodological rigor and refining their interpretation of the results, the authors can significantly enhance the impact and clarity of their contribution.

I hope these comments and suggestions are helpful in revising and strengthening your manuscript. I look forward to seeing the revised version of your work.

R2.5. We sincerely appreciate your thoughtful summary of recommendations and the constructive feedback provided. Incorporating these suggestions has greatly enhanced the rigor and clarity of our study.

Reviewer 3

3.1 Summary: This paper sets out to quantify how much occupations have changed in terms of their skill requirements between 2010 and 2018. To do so, it embeds skills in a high dimensional space using an approach from natural language processing (word2vec). The embedding captures information about how about 16k different skills are combined in the skill requirements of 167M job ads. The authors then use these embeddings to assess how entire occupations change. To do so, they average the embeddings of a set of core skills that are dominant in an occupation's job ads. Finally, they use this metric to explore which occupations changed the most over the period of their study. The main result shows that the use of skill embeddings overturns earlier findings that high-skill jobs had changed faster than low-skilled jobs. In fact, properly accounting for skill distances shows that the opposite holds: it is the low skill jobs that changed fastest. Furthermore, the authors show that jobs change faster in smaller labor markets and smaller employers. Finally, low-skilled jobs that are predominantly held by non-white workers experienced particularly significant up-skilling requirements.

R3.1. We are pleased that you clearly apprehend our research design.

3.2 Review: The paper is part of a growing literature that tries to understand the nature of jobs in terms of highly disaggregated task or skill requirements. Its main contribution is that it quantifies how jobs change in a very large dataset of job ads, offering a high level of detail and wide, up-to-date coverage. Its main methodological novelty lies in using word2vec algorithms to reduce the dimensionality of the skill space. The relevance of doing so is immediately apparent, because it provides a diametrically opposed answer to an important question compared to existing approaches that count the number of skills that change in each occupation: it is not the high-skill jobs that changed most, but the low-skill ones. This has important implications not only for policy, but also for individuals' career planning and hiring decisions by companies. Overall, the paper is carefully executed and includes many important checks, for instance, about the representivity of the sample and the performance of the identified skill spaces in predicting job switches. Therefore, I am in general very positive about this work. However, I have some concerns and suggestions that I believe should be addressed. I list these below.

R3.2. We greatly appreciate your positive assessment and recognition of our work's relevance, methodological novelty, and its important implications for policy, career planning, and hiring decisions. Your support highlights the potential impact of this research in both academia and practice. We have incorporated your suggestions, detailed below, to further strengthen the paper and address your thoughtful concerns. Thank you for supporting this work and recognizing its impact.

3.3 What are occupations? The authors ask by how much occupations change between 2010 and 2018. However, occupations are an artificial unit of observation that depends on the algorithm that the data provider used to classify jobs into occupations. To be sure, this problem is shared across the literature that relies on job ad data. However, given that the core research question is about within-occupation skill change it would be useful to reflect on the extent to which occupations are a useful unit of analysis. To explain better what I have in mind, there are two ways to think about dynamics on labor markets. One is careers, where workers move from one job to another. The other is vacancy chains, where jobs are moved from worker to worker. In the former, it is straightforward to determine how workers string jobs into a career. In the latter, it is much harder to decide which vacancies connect to the same underlying job and therewith to each other. The authors have data on vacancies, but are mostly interested in what happens to workers (for instance, which workers face the largest reskilling pressures). This leads to some complications.

First, even if job ads are unbiased reflections of vacancies, vacancies and jobs are not the same. For one, the vacancy-to-job ratio depends, among other things, on the turnover and average tenure in a job (jobs with much turnover will become vacant more frequently). This in turn is often related to average age and skill levels of workers. For another, it is not always clear whether the content of a job changes or whether an old job disappears and a new one appears. As a concrete example, consider the Gaming Dealers in Fig S3. In this occupation, skill requirements shifted from hospitality industry knowledge to big data processing. Arguably, however, this shift reflects two different jobs, one on the casino floor and the other in the IT department and it is unlikely that the same type of worker would be hired for these jobs or that a workers would transition between these task profiles (in fact, the old occupation's profile might have been closer to restaurant or hotel jobs than to the new occupation's profile). However, due to the data provider's classification, this is considered a within-occupation shift in skill requirements. To be sure, I don't think this paper is the place to resolve such ambiguities, but it would be helpful (1) to understand how job ads are classified into occupations by the data provider, (2) how the authors think about within versus across occupation skill changes and (3) reflect on the conceptual differences between job and vacancy data.

R3.3. Thank you for these sharp questions. We agree these points should be better clarified.

1. The data provider classified job ads into occupations with a proprietary process that combines machine learning (ML) models with rules curated by their in-house taxonomy team. Rules are applied first, and the model is used only if no rule matches. Here's the data provider's description of the process: "The classifier uses a job posting's title, description and regional information (i.e. origin country and language) in order to classify the content to one of Lightcast's Specialized Occupations. Taxonomists curate parts of the model to target the specific keywords, skills, and terminology of our Specialized Occupations. The remaining portion of the model learns from a corpus of raw job

postings data. This ensures that our occupation classifier is picking up on all the skills, relevant certifications, representative roles and responsibilities, and other industry or occupation specific terminology in order to classify the posting correctly. Common words, company boilerplates, and benefits are removed prior to classification.” See more at lightcast website: Job Posting Analytics (JPA) Methodology (https://kb.lightcast.io/en/articles/6957446-job-posting-analytics-jpa-methodology#h_cfb69c029c) and Lightcast Occupations Taxonomy (LOT) Classification Methodology (<https://kb.lightcast.io/en/articles/7907688-lightcast-occupations-taxonomy-lot-classification-methodology>).

2. First, Figure S6 (former Figure S3) is about skill atoms and not actual individual skill transformation, the IT skills gaming dealers actually developed are mostly database management softwares and platforms, such as apache hive, apache hadoop, apache kafka, and sql. On O*NET, the description of occupation “Gambling Dealer” listed Cloud-based data access and sharing software and Data base management system software under the technical skill requirement of this occupation, including examples of Apache Hadoop and Apache Spark. The O*NET job zone classification for this occupation is 2: suggesting it often requires a high-school diploma, and some level of experience and training. We hope such consistency with an external data source could alleviate the concern of systematic misclassification in our data. Nevertheless, we agree this pattern observed in our data and O*NET could possibly reflect a new, rather distinct job role creation under this occupation.

Second, our occupational skill change measure does reflect both shifts in within-job-role skill requirements and job role composition within an occupation. To more precisely disentangle the two, we construct a measure for job role composition changes with changes in within-occupational job role homogeneity level. The level of job role homogeneity is measured by the average pairwise job post skill similarity within an occupation in a given year. Changes in this metric reflect whether an occupation becomes more or less diverse in its job role composition by adding or shredding job roles that are very different from the main job roles. To single out skill changes from job role composition changes, we control for the changes in within-occupation job role homogeneity level in all the regressions (table S4-S9). Our findings still hold after adding this control.

Third, while an occupation could encompass multiple job roles that are not identical, we still find occupation an meaningful unit of analysis. Based on 2018 CPS-ASEC data, about 87% of observations (with non-missing occupation variables) stay in the same occupation as last year. This high level of occupational stability suggests that occupations

serve as a relatively consistent framework for analyzing individuals' roles in the labor market over time.

3. We acknowledge that our analysis is demand-driven rather than supply-driven, meaning the observed skill changes reflect re-skilling pressures under the assumption of a fixed supply, rather than actual re-skilling behavior. We would like to emphasize that vacancy data reflects shifts in employer expectations and provides valuable insights. We regret not making this clear in the previous version and have clarified this in the revised discussion.

We also acknowledge the distinctions between vacancies and incumbent jobs. As Reviewer #3 noted, the vacancy-to-incumbency ratio is influenced by turnover rates and average job tenure. Low-skilled jobs tend to have higher turnover and shorter tenures, leading to more frequent vacancies compared to high-skilled occupations. While we lack data to directly compare skill requirements between vacancies and incumbent jobs within the same occupation, one potential hypothesis could be that: vacancies may reflect greater skill changes than incumbent jobs, as it is easier to implement changes for future hires. Given that the vacancy-to-incumbency ratio is higher for low-skilled occupations, vacancy data may more accurately capture average skill changes (across both vacancies and incumbents) for these roles. In contrast, vacancy data for high-skilled occupations may overestimate average skill changes due to the larger share of more stable incumbent jobs. If this hypothesis holds, our results may underestimate the skill change gap between low-skilled and high-skilled occupations.

We have clarified the limitations of vacancy data and called for future research to examine skill requirements in incumbent jobs to complement vacancy-based analyses in the discussion section.

3.4 How does the paper's methodology compare to other work on occupational distance measures? In the introduction, the authors motivate their work in part with the following statement: "The limitation of [Deming and Noray's] quantification, however, lies in its inadequate consideration of the graduated relationship between different skills. This oversight results in a skewed perspective, inducing an upward bias for technology-intensive and specialized occupations that possess many more listed "skills" than occupations with less specialization." To analyze how occupations change over time, the authors quantify the distance between the occupations' old and new skill vectors. Whereas their focus on within-occupation skill requirements changes and the use of job ad data are rare, measures of skill distances have been widely used in labor economics (mostly, to study job switches). From this literature, the authors refer to the work by Gathmann and Schoenberg (2010 – Journal of Labor Economics). This paper shares the same problem as Deming and Noray's work that the authors are right to criticize: it assumes that all skills are equidistant. However, other papers in this literature address

this problem in a similar way as the authors do, namely, using dimensionality reduction techniques, albeit more traditional ones, such as factor analysis. For instance, Ingram and Neumann (2006 – Labour Economics) use factor analysis on skills/tasks listed in the Dictionary of Occupational titles to study the returns to different skills. The same approach is used by Poletaev and Robinson (2008, Journal of Labor Economics) to measure skill distance in job switches. Finally, Robinson (2018, Journal of Human Resources) and Neffke et al. (2024, Research Policy) use factor analysis and then calculate weighted differences between these lower dimensional skill vectors, where weights reflect the contribution of each skill factor to wages or years of schooling requirements to take into consideration how valuable skills are (Robinson) or how long it would take workers to acquire the required additional skills in a job switch (Neffke et al.). Note that the latter question is closely related to the analysis in Fig. S7 and Table S3 that aims to connect skill differences to required reskilling efforts, as well as to remarks about reskilling in the discussion section. Furthermore, these papers carry out similar analyses to the ones in appendix 3.5, but more in-depth than the current paper.

R3.4. Thank you for pointing this out, we have expanded the discussion section to build on these prior works, highlighting how they inform and connect to our method. We also emphasize the conceptual advantage of neural embedding over conventional dimension reduction methods. Our skill embedding approach offers a key advantage over traditional factor analysis by capturing both inter- and intra-categories of skill distance in a continuous space. Factor analysis, by contrast, assumes that skill categories (factors) are orthogonal, meaning there is no meaningful distance between them. As a result, when estimating skill transition difficulty, factor analysis must treat all factors and their associated skills as equally distant, ignoring the varying degrees of relatedness between skills. In contrast, our embedding approach learns a continuous skill space where distances between skills reflect their transition difficulty within and across categories. This allows for a more precise measurement of reskilling pressure, particularly for low-skilled jobs where skill shifts are more drastic across categories. In sum, while both methods may predict job transitions similarly, our approach provides richer insights into which workers face the greatest challenges in skill adaptation, making it a more actionable tool for workforce policy.

3.5 This shows that there is quite a bit of work on embedding skills in lower dimensional spaces to capture the “graduated relationship between different skills.” This does not mean that the use of more modern dimensionality reduction methodologies like word2vec is not valuable. However, to fully assess the value of these methods, one would have to compare them to existing efforts. In particular, I would have found it helpful to learn more about two issues. First, word2vec seems less transparent than factor analysis and it would be helpful to understand exactly what kind of information is used to measure how similar two skills are. Currently, the authors state: “This embedding space encodes the similarity and distance between skills inferred from their direct and indirect co-presence across jobs.” What is conceptually the meaning of a direct co-occurrence? What of an indirect co-occurrence? In what sense are skills that co-occur

directly or indirectly similar? How much weight is put on either type of co-occurrence? If the last aspect is hard to quantify, it would already be helpful to answer the first questions to better understand what is going on under the hood. Second, it would be helpful to learn if word2vec based differences are more predictive of, for instance, job switches (as analyzed in the section titled: Validating Skill Vectors with Worker Mobility) than the factor-analysis based measures proposed in the labor literature. For instance, if factor analysis is used to reduce dimensionality how much of the variance in job switching does the Euclidian or cosine distance between two occupations in this space explain? Establishing that the authors' approach performs better would help its adoption in future work.

R3.5. We have sought to address these two issues:

1. We now articulate the meaning of a direct and indirect co-occurrence in the methods section. A direct co-occurrence between two skills manifests when they appear in a job advertisement together. A first-order indirect co-occurrence between two skills manifests when another co-occurs with each skill, creating an indirect skill co-occurrence path. Further indirectness occurs when skills connect the two skills along more distant pathways of co-occurrence. This association of directness and indirectness is akin to the direct and indirect pathways through which spectral algorithms underlying PageRank (Bianchini, Gori, and Scarselli 2005) and other Eigenvalue-based centrality scores (Bonacich 1987) account for direct centrality, where other nodes connect to a focal node, and indirect centrality, where other nodes connect to nodes that themselves connect with the focal node. Direct co-occurrence disproportionately influences our assessments as the resulting neural embedding proximity approximates the integration of all co-occurrence pathways between skills, and direct co-occurrence represents a much more probable pathway between two skills than indirect co-occurrence.
2. We have compared word2vec with factor analysis both theoretically (in the discussion section) and in the application to measure job differences (in the methods section and SI section 2.6). With 200 latent factors learned from the weighted occupation-skill probability matrix, the Euclidean distance or cosine similarity between occupation pairs represented by factors are highly correlated with the distances calculated based on skill embedding representation of occupations: the Pearson correlation is 0.51 ($p \sim 0$) for Euclidean distance and 0.59 ($p \sim 0$) for cosine similarity. The two approaches also perform similarly in predicting job switches, with skill2vec explaining slightly more variance in job switches. We found that skill2vec-based distances outperform factor-based occupation-pair distances, regardless of how distance is defined. The R^2 for regression on job transitions increased by 37% (from 0.095 to 0.131) when using Euclidean distance and by 3% (from 0.137 to 0.141) when using cosine similarity.

3.6 The regression analysis in Tables 5.4-5.7 strikes me as odd. This analysis tries to assess whether occupation change is more rapid in large or small employers and in large or small labor markets. While the question is interesting, I have major concerns about how the sample is created. The sample now consists, as far as I understood, of 488 occupations in four different contexts: (1) large employers (2) small employers (3) large labor markets (4) small labor markets, yielding a sample with $488 \times 4 = 1952$ observations. However, what is this supposed to be a sample of? After all, regression analysis is supposed to uncover the conditional expectation of a dependent variable as a function of a set of independent variables. The sample should be a reflection of some population from which units are drawn, ideally at random, such that this expectation can be estimated. However, what is the population of occupation-context cells supposed to reflect? This is conceptually hard to justify and in practice ignores that some cells are much larger (and therewith yield more accurate averages) than other cells.

Moreover, occupational changes in such cells may be due to actual changes of jobs, but also a shift of jobs between employers that prefer one version of an occupation to employers that prefer another. Note that the authors motivate the regression analysis as follows on page 4: “On one hand, research suggests that job roles in large firms and markets may change faster. These entities are often leading their fields at the skill frontier and possess the resources to attract and retain highly skilled workers. [...] On the other hand, smaller firms and markets could pursue more significant skill upgrading to catch up with their larger counterparts.” Clearly, these hypotheses reflect considerations about changing jobs within a firm, but the regression analysis is not set up to capture such changes well and will mix within and cross-firm changes.

Instead, the authors could focus on more intuitive statistical units (e.g., firms). This is hard, because the unit of analysis would ideally be a job in a firm (see point 1 above), which requires connecting job ads in 2010 to job ads in 2018 within the same firm. Alternatively, one could study at the occupation-firm level and employ proper weighting to reflect that some occupations are larger than others. This still leaves a problem of appearing and disappearing occupations. However, I would consider this a feature of the world, not a problem of the methodology. To account for this, the authors could study how the firm changes as a whole (i.e., averaging skill vectors of all job ads related to the same firm, once in 2010 and once in 2018) and compare this to how occupations within firms change. A similar solution could exist for appearing and disappearing firms, whose skill vectors can be compared to one another. Note that adding information on the size of the labor market is straightforward, because either firms only have one location or the authors could split multilocation firms.

R3.6. We have revised the regressions with more meaningful statistical units and proper weighting to reflect different sizes of the unit cells as suggested:

1. Table S4 and S7 contain regressions with an occupation as an analytical unit. Each occupation is weighted by the average number of job posts in 2010 and 2018 to adjust for the fact that some occupation cells are larger and have more accurate averages.

2. Table S5 and S8 contain regressions with a firm-commuting zone-occupation as an analytical unit to track within-firm-occupation changes, using a sample of units that appear in both 2010 and 2018. Each firm-commuting zone-occupation unit is weighted by the corresponding average number of job posts in 2010 and 2018.
3. Table S6 and S9 contain regressions with a firm-commuting zone as an analytical unit to include occupations that disappeared or newly appeared in 2018, using a sample of units that appear in both 2010 and 2018. Each firm-commuting zone is weighted by the corresponding average number of job posts in 2010 and 2018.

3.7 Minor comments: a. In the analysis of demographics, could the authors give more information on what kind of occupations they focus on? Typically, extreme overrepresentation is easier in small occupations than in larger ones. Moreover, it would be equally easy to just calculate weighted skill changes using a group's distribution across occupations as weights. In a sense, this would be a more accurate reflection of the types of skill changes different societal groups face. Does this change results?

R3.7. Thanks for pointing out the potential bias in our approach to assessing skill change faced by different social groups. We really appreciate the suggested more accurate and intuitive approach and used it in place of the original approach in demographic analysis and Figure 3g. For each social group, we calculate the weighted sum of occupational skill changes across all occupations, where the weights correspond to the proportion of the group's workers employed in each occupation relative to the group's total employment across occupations. With this measurement, we find that non-white workers change more than white workers, and female workers change more than male workers.

3.8 In the discussion, the costs of relocation mentioned are transportation and moving costs. However, it is likely that the true costs of relocations are nonpecuniary, such as losses of the social capital and networks that support individuals both at a professional and a personal level.

R3.8. We agree with this insightful point and have added it in the discussion of the relocation costs.

3.9 I did not fully follow the following sentence on page 23 "To account for temporal compositional changes in job post number and skill number per job post, they weight the skill change rate calculated from equation (1) by multiplying the ratio of skill occurrence divided by post number in 2007 to that in 2019, for each occupation." Adding an equation for these weights would be helpful.

R3.9. We added an equation for these weights to equation (1). These weights are used to prevent confusing changes in job posting numbers with changes in the number of skills per vacancy. To

clarify, for each occupation o , the weight consists of the numerator, the ratio of the 2007 required skill number to the 2019 required skill number, divided by the denominator, the ratio of the 2007 job post number to the 2019 job post number.

3.10 Notation: the subscript o in the righthand side of equation 6, page 26 turns into a variable (O) on the lefthand side, which is confusing.

R3.10. Yes. We now change the notations of the variable on the left hand side to (L) to avoid confusion.

3.11 How much variance is there within occupations in skill vectors? Are some occupations much more precisely defined than others? What would that mean?

R3.11. We constructed a measure of within-occupation job role homogeneity/variance level by calculating the average pairwise job post-skill similarity within an occupation in a given year. A higher score means less within-occupation variance: a score of 1 means all job posts are completely identical in skill contents. The average occupational homogeneity score based on 2010 and 2018 data is 0.63 (standard deviation 0.11). Some occupations have higher job role homogeneity than others, meaning that they have more consistent skill requirements across all job posts. Occupations that are internally more homogeneous might be in more stable job fields, embedded in more specific contexts, and therefore are prone to have less dramatic skill changes. Therefore, we control for occupational job role homogeneity in the regressions to rule out any potential confounding effect – the results are robust to this control.

3.12 Overall, I enjoyed reading the paper and I believe that its conclusions are sound. I hope these comments will be helpful to the authors.

R3.12. We have incorporated your suggestions to further strengthen the paper and address your thoughtful concerns. Thank you for supporting this work and recognizing its impact.

Bianchini, Monica, Marco Gori, and Franco Scarselli. 2005. "Inside PageRank." *ACM Transactions on Internet Technology* 5 (1): 92–128.

Bonacich, Phillip. 1987. "Power and Centrality: A Family of Measures." *The American Journal of Sociology* 92 (5): 1170–82.

Dear Editors and Reviewers,

We thank you for the encouraging assessment and thoughtful feedback. As the reviewers note, only minor issues remain, which we have now addressed through improved organization of the discussion, clarification of the supplementary materials, and correction of typos in equations and the main text. We believe these final refinements strengthen the manuscript's clarity and rigor, and we hope it is now ready for publication at *Nature Communications*. Below, we provide a point-by-point response to each comment.

Reviewer 1

The authors have satisfactorily addressed my previous comments.

R1.1. We are pleased to hear that you found our revisions satisfactory. Thank you for your time and constructive feedback.

Reviewer 2

The authors have thoroughly and thoughtfully addressed my initial concerns, substantially improving both the methodological rigor and interpretative clarity of the manuscript. They have revised their framing to provide a more nuanced discussion of the observed skill changes, appropriately distinguishing between IT skill diffusion and broader occupational upskilling. Furthermore, they implemented robust validation procedures for the embedding space, including comparisons with a state-of-the-art embedding model, visualization of skill and occupational vectors, and enhanced methodological transparency. I sincerely thank the authors for their careful and diligent revisions and for respectfully engaging with my comments. These thoughtful efforts have significantly strengthened the study. I believe the current version is well-prepared for publication in *Nature Communications*.

R2.1. We are truly grateful for your generous and thoughtful review. Your comments played a vital role in improving both the methodological rigor and interpretative clarity of the manuscript. We especially appreciate your recognition of our efforts to refine the framing, clarify the distinction between IT skill diffusion and broader upskilling, and enhance the transparency and robustness of our validation procedures. Thank you again for your constructive feedback and encouragement.

Reviewer 3

3.1 The authors substantially revised and expanded the initial version of the paper. In doing so, they addressed all my main concerns. I believe the current version is convincing and makes an

important contribution to our understanding of changes in the US labor market in terms of the detailed skills that jobs require. I have a few minor points that the authors should be able to address with little effort and that would amount to minor revisions:

R3.1. Thank you for your generous and constructive review. We are glad to hear that you find the revised manuscript convincing and valuable. We appreciate your remaining suggestions and have addressed each of them carefully below.

3.2 1) Supplementary material, section 2.1. a. The authors should define each of the variables in these equations. b. Eq (1b) is o -specific, but the right-hand side variables contain no subscript o . c. I think eq. (1b) is the (inverse of the) growth rate of the average number of skills per job in an occupation (after rearranging terms). Is that correct? Or should t_0 and t_1 be reversed? As far as I understand, this just loosely weights occupations that grow in complexity (i.e., that add skills) less heavily than those that don't but it is no "principled" correction against some kind of null model, correct? Given that the authors need to replicate the paper that motivated their analysis and thus "inherit" this equation, I would not expect a justification for this correction (in fact, the original paper neither seems to give a clear definition nor explanation for these weights), but it would help to understand how we should think about them.

R3.2. Thank you for these helpful clarifications. We have now defined all variables in Supplementary Material Section 2.1 and added the missing subscript o to the right-hand side of Eq. (1b). You are correct—Eq. (1b) approximates the inverse growth rate of the average number of skills per job in each occupation. We have now clarified this interpretation in the text and confirmed the direction of t_0 and t_1 . As you correctly noted, this weighting follows the exact approach of Deming and Noray's (2020) study that we replicate, and is not a "principled" correction against some kind of null model. We now make this clearer in the revised supplementary note.

3.3 2) Supplementary material, section 2.6: wording. I would typically not interpret a Pearson correlation of .51 or .59 as high. These metrics are statistically associated, but far from the same. What is more important is that the R^2 in job transition predictions is higher for the word2vec embeddings, suggesting that the word2vec approach captures more relevant information.

R3.3. Thank you for this important clarification. We have revised the wording in Supplementary Material Section 2.6 to avoid describing the Pearson correlations (0.52 and 0.59) as "high." Instead, we now characterize them as moderate but statistically significant, and we emphasize the more important finding: that the word2vec-based distances explain more variance in job transitions than the factor-based approach.

3.4 3) Potential inconsistencies about the relation between unemployment rate and skill change.

a. In the main text, the authors claim “we show evidence consistent with prior research that lower employment rates are associated with smaller yearly average upskilling change, 2010 to 2018.” There, the authors discuss employment rates, which move opposite of unemployment rates. b. In the SI, section 3.5, the authors claim: “We computed the year-on-year average occupational skill changes (e.g., 2010-2011, 2011-2012, etc.) across all occupations from 2010-2018 and found a positive correlation with the yearly civilian unemployment rate released by BLS: pearson coefficient 0.95 with p-value less than 0.005. Fig. S11 shows that the yearly average occupational skill changes decline with the yearly unemployment rate from 2010-2018. Given that our skill change measure largely captures upskilling directions, this pattern is consistent with the business cycle literature.” Here, the stated correlation is positive, suggesting that the correlation with employment rates should be negative (i.e., opposite of what the main text claims). In line with this, Fig. 11 suggests that skill comove with unemployment rates. Is there a typo in the main text?

R3.4. Thank you for catching this inconsistency. You are absolutely right—the main text contains a typo. We intended to write “lower *unemployment* rates” (not *employment* rates) when summarizing the results from Supplementary Material Section 3.5. We have corrected this in the revised main text to accurately reflect the pattern shown in Fig. S11 and the business cycle literature.

3.5 4) Section 3.6 of the SI. The regression analysis is now much more interpretable. However, it seems that the authors use the number of ads in a cell as weights. That is, they use frequency weights. The correct weights would be analytic weights. Can the authors clarify which weights they use?

R3.5. Thank you for this helpful clarification. In the original regressions, we used frequency weights (i.e., duplicating rows) based on the number of job ads in each unit (occupation, firm-cz-occupation, or firm-cz) to give more weight to estimates based on larger underlying samples. Following your suggestion, we reran the analyses using analytic weights via Weighted Least Squares (WLS) to account for varying precision across units. The results remain robust, with comparable effect sizes and significance levels. We have updated regression tables in Supplementary Section 3.6 and 3.7.

3.6 5) I found the discussion section hard to process, because it does not clearly group different types of arguments. Currently, the section mixes summary, limitations, new discussions of other papers and even some additional results to conclude with a somewhat generic call for policy based on the paper’s results. I think the paper could be made more impactful by improving the structure and flow of this section, clearly separating findings, limitations and open questions.

R3.6. Thank you for this valuable suggestion. In response, we have rewritten the Discussion section to improve clarity and flow. The revised version now explicitly separates the key findings, limitations, and open questions into distinct paragraphs. We have also moved new empirical details to earlier sections where appropriate and refined the concluding policy discussion to better reflect the implications of our core results. We believe these changes make the narrative more coherent and the paper's contributions more impactful.